# The BLM-TOP3A-RMI1-RMI2 proximity map reveals that RAD54L2 suppresses sister chromatid exchanges

Jung Jennifer Ho[1,2], Edith Cheng[1,2], Cassandra J Wong[3], Jonathan R St-Germain[4,5], Wade H Dunham[3], Brian Raught[4,5], Anne-Claude Gingras [ID] [3,6] & Grant W Brown [ID] [1,2✉]

## Abstract

Homologous recombination is a largely error-free DNA repair mechanism conserved across all domains of life and is essential for the maintenance of genome integrity. Not only are the mutations in homologous recombination repair genes probable cancer drivers, some also cause genetic disorders. In particular, mutations in the Bloom (BLM) helicase cause Bloom Syndrome, a rare autosomal recessive disorder characterized by increased sister chromatid exchanges and predisposition to a variety of cancers. The pathology of Bloom Syndrome stems from the impaired activity of the BLM-TOP3A-RMI1-RMI2 (BTRR) complex which suppresses crossover recombination to prevent potentially deleterious genome rearrangements. We provide a comprehensive BTRR proximal proteome, revealing proteins that suppress crossover recombination. We find that RAD54L2, a SNF2-family protein, physically interacts with BLM and suppresses sister chromatid exchanges. RAD54L2 is important for recruitment of BLM to chromatin and requires an intact ATPase domain to promote non-crossover recombination. Thus, the BTRR proximity map identifies a regulator of recombination.

Keywords Recombination; Sister Chromatid Exchanges; Bloom Syndrome; BioID; RAD54L2
Subject Category DNA Replication, Recombination & Repair

## Introduction

The repair of double-strand DNA breaks (DSBs) by homologous recombination (HR) is critical for the maintenance of genome stability. HR is a high-fidelity repair process that employs homologous DNA sequences, typically in the sister chromatid, as a template to repair DSBs while retaining critical sequence information. Mutations in HR genes can cause genome instability and predisposition to cancer; for example, ataxia telangiectasia, Nijmegen breakage syndrome, and Bloom syndrome are caused by mutations in ATM, NBN, and BLM, respectively, and are associated with pleiotropic cancer susceptibility (Renwick et al, 2006; Belhadj et al, 2023; German, 1997). Bloom syndrome is a rare disease characterized by small stature, extreme sensitivity to sunlight, immunodeficiency, and oncogenesis (reviewed in (de Renty and Ellis, 2017)). The DNA helicase BLM plays a critical role in DSB repair to suppress oncogenesis. BLM functions in a complex with the topoisomerase TOP3A (Hu et al, 2001; Wu et al, 2000; Johnson et al, 2000) and the structural components RMI1 (Meetei et al, 2003; Yin et al, 2005; Raynard et al, 2006) and RMI2 (Singh et al, 2008; Xu et al, 2008) to suppress rearrangements during HR repair of DSBs (Bythell-Douglas and Deans, 2021) and stressed DNA replication forks (Lönn et al, 1990; Sengupta et al, 2003; Davies et al, 2004, 2007), thereby maintaining genome integrity.

BLM responds to the presence of DSBs both early and late in HR repair. In the early stages of repair, BLM DNA helicase activity promotes the long-range resection of double-stranded DNA ends in concert with the DNA2 endonuclease (Gravel et al, 2008; Nimonkar et al, 2011). The resulting 3′ single-stranded DNA tails are substrates for the assembly of RAD51-ssDNA filaments, which conduct homology search for suitable repair templates (Zhao et al, 2015). In the late stages of HR repair, BLM functions in a complex with TOP3A (Hu et al, 2001; Wu et al, 2000; Hodson et al, 2022), RMI1 (Yin et al, 2005; Wu et al, 2006), and RMI2 (Singh et al, 2008) to dissolve four-way recombination intermediates known as double Holliday junctions (dHJs) (Bizard and Hickson, 2014). Dissolution of dHJs by the BLM-TOP3A-RMI1-RMI2 (BTRR) complex is distinguished from dHJ resolution by structure-selective nucleases in that dissolution produces exclusively noncrossover products. As such, defects in BTRR components result in the canonical phenotype of BLM syndrome: increased sister chromatid exchanges (SCEs) (German et al, 1977; Martin et al, 2018; Gönenc et al, 2022; Hudson et al, 2016), increased loss of heterozygosity events (Yusa et al, 2004; Langlois et al, 1989; LaRocque et al, 2011), and increased risks of pleiotropic cancer (German, 1997; Ababou,

[1]Department of Biochemistry, University of Toronto, 1 King's College Circle, Toronto, ON M5S 1A8, Canada. [2]Donnelly Centre for Cellular and Biomolecular Research, University of Toronto, 160 College Street, Toronto, ON M5S 3E1, Canada. [3]Lunenfeld-Tanenbaum Research Institute, Mount Sinai Hospital, Sinai Health, Toronto, ON, Canada. [4]Princess Margaret Cancer Centre, University Health Network, Toronto, ON, Canada. [5]Department of Medical Biophysics, University of Toronto, Toronto, ON, Canada. [6]Department of Molecular Genetics, University of Toronto, Toronto, ON, Canada. ✉E-mail: grant.brown@utoronto.ca

2021). Elevated SCEs are a broad indicator of genome instability and occur genome-wide (Hamadeh et al, 2022) and, in particular, in hotspots at common fragile sites (van Wietmarschen et al, 2018). SCEs are associated with the use of therapeutics that stall DNA replication such as PARP inhibitors (Heijink et al, 2022), and those that generate DSBs, such as irradiation (Conrad et al, 2011).

In addition to its primary function in the suppression of crossover recombination, BLM functions in many other aspects of genome stability. These include the restart of stalled replication forks (Davies et al, 2007), unwinding of RNA G-quadruplexes in stress granules (Danino et al, 2023), unwinding of DNA G-quadruplexes during telomere replication (Drosopoulos et al, 2015), and the repair of ultra-fine DNA bridges in anaphase (Chan et al, 2007). It is thought that the BLM function is largely defined by its interacting partners and nucleic acid substrates. For example, RPA binding is important for BLM function in replication fork restart but not in recombination or ultra-fine DNA bridge processing (Shorrocks et al, 2021).

BLM-interacting proteins have been identified by several affinity-purification/mass spectrometry approaches, including affinity purification of streptavidin-tagged BLM (Wan et al, 2013), immunoprecipitation of endogenous BLM using anti-BLM antibodies (Yin et al, 2005; Meetei et al, 2003; Guo et al, 2023; Bhattacharyya et al, 2009), and affinity purification of engineered fluorescent protein BLM fusions (Hein et al, 2015; Cho et al, 2022), each coupled to mass spectrometry to identify interacting proteins. Despite these comprehensive affinity-purification interactomes, BTRR is often found in nuclear foci (Wang et al, 2022; Eladad et al, 2005; Davalos et al, 2004), which are biochemically unstable and insoluble (Takata et al, 2009), making it challenging to capture a complete and physiologically relevant BTRR interactome.

Here, we map the BTRR proximal proteome using proximity-dependent biotin identification (BioID) and identify proteins that suppress crossover recombination. We define 206 high-confidence BTRR proximal proteins, including 24 proximal proteins that are shared by at least two BTRR subunits. Comparison of N- versus C-terminal BLM BioID suggests most BLM interactions occur in proximity to the N-terminus. Analysis of the BTRR proximal proteome revealed 13 proteins that suppress SCEs, including the SNF2-family ATPase RAD54L2. Knockout of *RAD54L2* results in increased SCEs and decreased noncrossover recombination. RAD54L2 is important for recruitment of BLM to repair foci, but does not influence MRE11 or RAD51 recruitment, indicating that RAD54L2 likely functions during dissolution of dHJs. Thus, we identify a new player in the processing of recombination intermediates.

## Results

### The BTRR proximal proteome

To capture the proximal proteome of the BTRR complex, we performed BioID to selectively biotinylate proteins in close proximity to each BTRR complex member (Roux et al, 2012). We generated four tetracycline-inducible HEK293 cell lines that stably expressed N-terminal BirA* tagged BTRR fusion proteins (Fig. EV1). Following 24-h induction of each BirA* fusion protein,

biotinylated proteins were affinity-purified and analyzed by mass spectrometry. The confidence of each BTRR proximity interaction was assessed with SAINTexpress (Teo et al, 2014) to predict the likelihood of a true interaction for each bait-prey pair. We detected a total of 64 high-confidence proximal proteins (Bayesian false discovery rate (BFDR) ≤ 0.01) for BLM, 18 for TOP3A, 84 for RMI1, and 40 for RMI2 (Fig. 1A,B).

Previous analyses have detected diverse proteins physically interacting, directly or indirectly, with BTRR. Comparing to protein interactors with low-throughput evidence in the BioGRID database (Oughtred et al, 2021) (https://thebiogrid.org/; accessed 26/09/2023; Source Data 1A,B), we detected 15 of 78 annotated interactions for BLM, 3 of 18 for TOP3A, 6 of 10 for RMI1, and 3 of 8 for RMI2 (Fig. 1A,B and Source Data 1A,B). Notably, the known interacting proteins of BTRR that we detected with BioID include components of the Fanconi anemia (FA) complex (FANCM, FANCD2, BRIP1/FANCJ, and CENPX) involved in double-strand break repair and DNA replication stress response (Meetei et al, 2003), double-strand break repair proteins (RAD50 (Franchitto and Pichierri, 2002) and BRCA1 (Acharya et al, 2014)), and proteins that repair stalled replication forks (TOPBP1 (Wang et al, 2013a), ETAA1 (Bass et al, 2016), WRN (Sturzenegger et al, 2014)). Thus, the proximal proteome of BTRR contains the expected network of DNA damage and replication stress response and repair proteins.

Twenty-four proximal proteins were detected in at least two BTRR BioID screens (Fig. 1A,C and Source Data 1A,C), 13 of which are not currently annotated as BTRR interactors in BioGRID. The known BTRR binding partner FANCM (Deans and West, 2009; Lu et al, 2019) was identified in all four screens. Interaction with FAAP24, an additional member of the FA complex (Ciccia et al, 2007; Coulthard et al, 2013) not currently known to interact with BTRR, was identified in three screens, as was the FA complex interactor CENPX. Analysis in GeneMANIA (Franz et al, 2018) (https://genemania.org/; accessed 24/01/2024) revealed an interacting network of proteins that were identified in at least two screens, including the tumor suppressor TP53, the p53 regulator PML, the DNA damage responsive E3 ligases UBR5 and ZNF451, and the DNA repair proteins BRCA1, CENPX, TOP2B, RPA1, and RPA2 (Fig. 1C and Source Data 1C).

We identified a total of 167 interactions that are not currently annotated in either low-throughput or high-throughput datasets (Oughtred et al, 2021). We performed a gene ontology analysis of the corresponding 150 genes, discovering statistically supported enrichments for DNA repair, DNA damage response, chromosome organization genes, and an unexpected enrichment for genes involved in non-membrane-bounded organelle assembly particularly genes involved in kinetochore and centrosome function (Fig. 1D and Source Data1D).

### The BLM N- and C-terminus proximal proteome

Since BLM is a 159 kDa multi-domain protein that binds protein partners via both N- and C-terminal regions (Yin et al, 2005; Wan et al, 2013; Meetei et al, 2003; Guo et al, 2023; Bhattacharyya et al, 2009), we aimed to define whether fusing a biotin ligase at the N- or C-terminus would reveal different proximal proteomes. We, therefore, fused the biotin ligase miniTurbo (mT) (Branon et al, 2018) to the N- or the C-terminus of BLM (Fig. 2). SAINTexpress (Teo et al, 2014) analysis of the miniTurbo proximity-labeling data

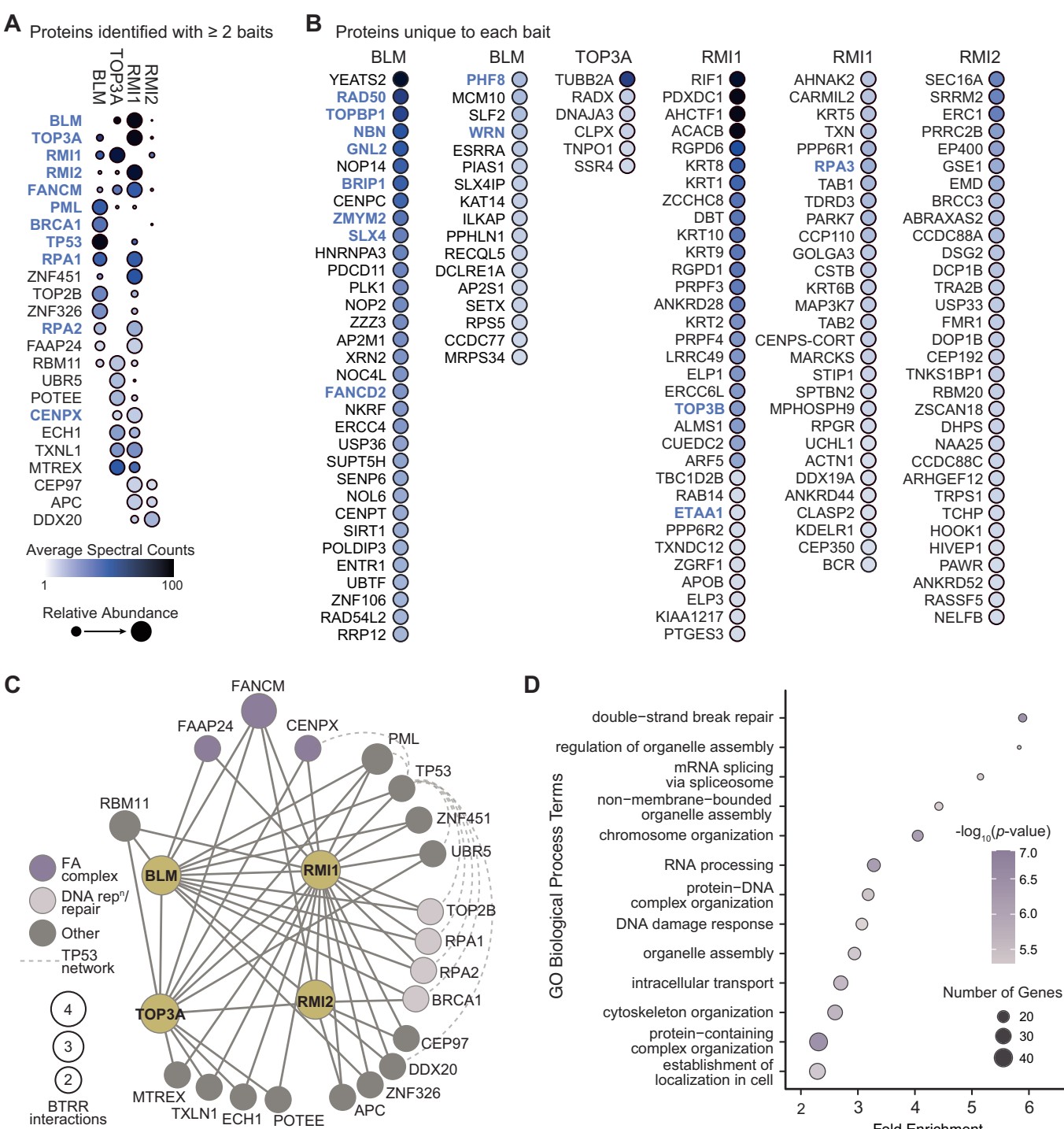

revealed 51 high-confidence proximity interactions with the N-terminal BLM fusion (mT-BLM) and 15 interactions with the C-terminal BLM fusion (BLM-mT) with BFDR ≤0.01 (Fig. 2A, Source Data 2A). Most of the interactions were detected with the N-terminal fusion, consistent with most of the annotated BLM interactions occurring via the amino terminus (Yin et al, 2005; Wan et al, 2013; Meetei et al, 2003; Guo et al, 2023; Bhattacharyya et al, 2009; Yankiwski et al, 2001).

We next combined the data from the three BLM BioID screens (BirA*-BLM, mT-BLM, and BLM-mT; Fig. 2B, Source Data 1AB, and Source Data 2B). The BLM proximal proteome comprises 101 high-confidence proximal proteins, and is highly enriched for annotated BLM protein–protein interactions (Oughtred et al, 2021) (27-fold; hypergeometric $p = 1.3 \times 10^{-30}$). Of note, we identified 76 proximal proteins that are not currently annotated, expanding the BLM interactome to 259 proteins. Functional analysis of the BLM

Figure 1.   BioID identifies proteins proximal to BLM, TOP3A, RMI1, and RMI2.

(A) Dot plot showing prey proteins identified with BirA*-BLM, BirA*- TOP3A, BirA*-RMI1, and BirA*-RMI2. High-confidence proximal proteins identified with at least two baits are shown (BFDR ≤0.01). BioID screens were performed in duplicate (BLM and TOP3A), triplicate (RMI2), or sextuplicate (RMI1). Dot color indicates the average spectral counts, and dot size indicates relative abundance across the baits. Previously described interaction partners annotated in BioGRID are shown in blue. (B) Dot plot showing all high-confidence prey proteins identified with one BTRR bait, with BFDR ≤0.01. Details as in panel (A). (C) Network of the 24 proximal proteins detected in at least two BTRR BioID screens. The nodes identify prey proteins and are sized according to the number of proximity interactions. The solid edges connect the BioID BTRR baits with their proximal prey. The dashed edges indicate protein–protein interactions among a subset of the BTRR proximal proteins as annotated in BioGRID. Members of the Fanconi anemia (FA) core complex, DNA replication/repair proteins, and a TP53 interaction network are indicated. (D) Gene ontology (GO) biological process analysis for 150 high-confidence BTRR proximal proteins not currently annotated in BioGRID. The -fold enrichment for each GO term is indicated, the colors indicate the corrected p-values, and the size of the circles corresponds to the number of genes annotated to the given GO term. The p-values were calculated using the hypergeometric distribution with Bonferroni multiple hypothesis correction. Source data are available online for this figure.

proximal proteome (Fig. 2C and Source Data 2C) indicated the expected enrichment for DNA recombination and repair, and revealed extensive interactions with post-translational modification pathways including sumoylation and acetylation.

## Proximal partners of BTRR suppress sister chromatid exchanges

To identify the high-confidence proximity interactions that are most relevant to BTRR function, we selected a subset of proximal proteins from our BTRR BioID screens for sister chromatid exchange (SCE) analysis. An increase in SCEs is the hallmark cellular phenotype of Bloom syndrome cells (German et al, 1977), and increased SCEs are evident when any component of BTRR is depleted (Martin et al, 2018; Hudson et al, 2016; Hoadley et al, 2010). We selected 18 genes from the BTRR proximal proteome to knock down with siRNA, and included BLM, TOP3A, and RMI1 as positive controls. Following siRNA depletion of each gene transcript, we stained the sister chromatids and prepared metaphase chromosome spreads (Fig. 3A). We assessed the effect of siRNA depletion of each candidate interactor on the number of SCEs per mitosis (Figs. 3B and EV3A; Source Data 3B). As expected, U2OS cells depleted of BLM or its known interacting partner, RMI1 showed increased frequency of SCEs. The siRNA depletion of TOP3A had a modest effect on SCEs (Figs. 3B and EV3A). Increased SCEs were evident for many candidate genes, encouraging more detailed analyses. In particular, 13 of the 18 gene knockdowns showed a greater than twofold increase in SCEs (Fig. 3B). Depletion of each individual protein was not assessed, so it is possible that the remaining 5 genes were false negatives in the SCE assay. We performed additional replicates to validate three candidates that exhibited high frequencies of SCEs (Figs. 3C and EV3B). Knockdown of RAD54L2, ABRAXAS2, or ZZZ3 using siRNA caused an increase in SCEs. The reciprocal of an SCE is noncrossover (NCO) recombination. We used a direct-repeat recombination assay (Weinstock et al, 2006), where NCOs reconstitute GFP after a DSB is induced, to assess NCO recombination following siRNA knockdown of RAD54L2, ABRAXAS2, or ZZZ3 (Figs. 3D and EV3C). Knockdown of each of the three genes resulted in decreased NCO recombination, consistent with the increase in SCEs we found upon knockdown. We infer that RAD54L2, ABRAXAS2, and ZZZ3 are important for suppressing crossovers following DSBs. RAD54L2 encodes an SNF2-family ATPase with sequence similarity to the recombination modulator RADX (Rouleau et al, 2002), and so we chose RAD54L2 for further analysis.

## RAD54L2 knockout increases crossover recombination

To confirm the role of RAD54L2 in suppressing crossovers we used CRISPR/Cas9 to disrupt RAD54L2 in U2OS cells. We tested two independent gene disruption lines for the frequency of SCEs (Figs. 3E and EV4A). In both cases, the number of SCEs per mitosis increased when RAD54L2 was disrupted. To gain additional insight into the function of RAD54L2 relative to that of BLM, we knocked down BLM in the RAD54L2 deficient lines and measured SCEs (Figs. 3E and EV4B). In both RAD54L2 deficient lines, knockdown of BLM increased the number of SCEs. However, the increase in SCEs was less than additive and less than multiplicative relative to the effect of single BLM knockdown or RAD54L2 knockout, consistent with BLM and RAD54L2 functioning at least partially in the same genetic pathway to suppress crossover events. We also noted (Fig. EV4B) that the steady-state level of BLM increased in the RAD54L2 knockout lines, indicating that the RAD54L2 knockout phenotype was not due to reduced BLM abundance. The reason for the increase in BLM levels is currently unclear.

We tested the role of RAD54L2 in noncrossover recombination by disrupting RAD54L2 in the NCO assay cell line (Figs. 3F and EV4C). Disruption of RAD54L2 in two different monoclonal lines resulted in a decrease in NCO events, almost to the extent seen with BLM knockdown. We introduced a wild-type copy of RAD54L2 into the RAD54L2 deficient lines, which rescued NCO events to wild-type levels (Fig. 3F and EV4D,E). Finally, we knocked down BLM in the RAD54L2 deficient lines and found that NCO events further decreased, to levels similar to those observed in BLM knockdown. We infer that BLM and RAD54L2 function in the same genetic pathway to promote NCO recombination repair outcomes.

## RAD54L2 is proximal to DNA repair proteins and chromatin regulators

To complement our mass spectrometry data, we tested whether we could detect RAD54L2 among the proteins in miniTurbo-BLM (Fig. 4A). RAD54L2 was detected in streptavidin precipitates only when miniTurbo-BLM was expressed, confirming the identification of RAD54L2 in BLM BioID experiments. We then asked if BLM immunoprecipitates contain RAD54L2 (Fig. 4B,C). Using our BioID cell lines, which express BLM fusion proteins when induced with doxycycline, we immunoprecipitated BirA*FLAG-BLM from nuclear extracts with antibodies against the FLAG epitope and probed immunoblots for BLM and RAD54L2. RAD54L2 was

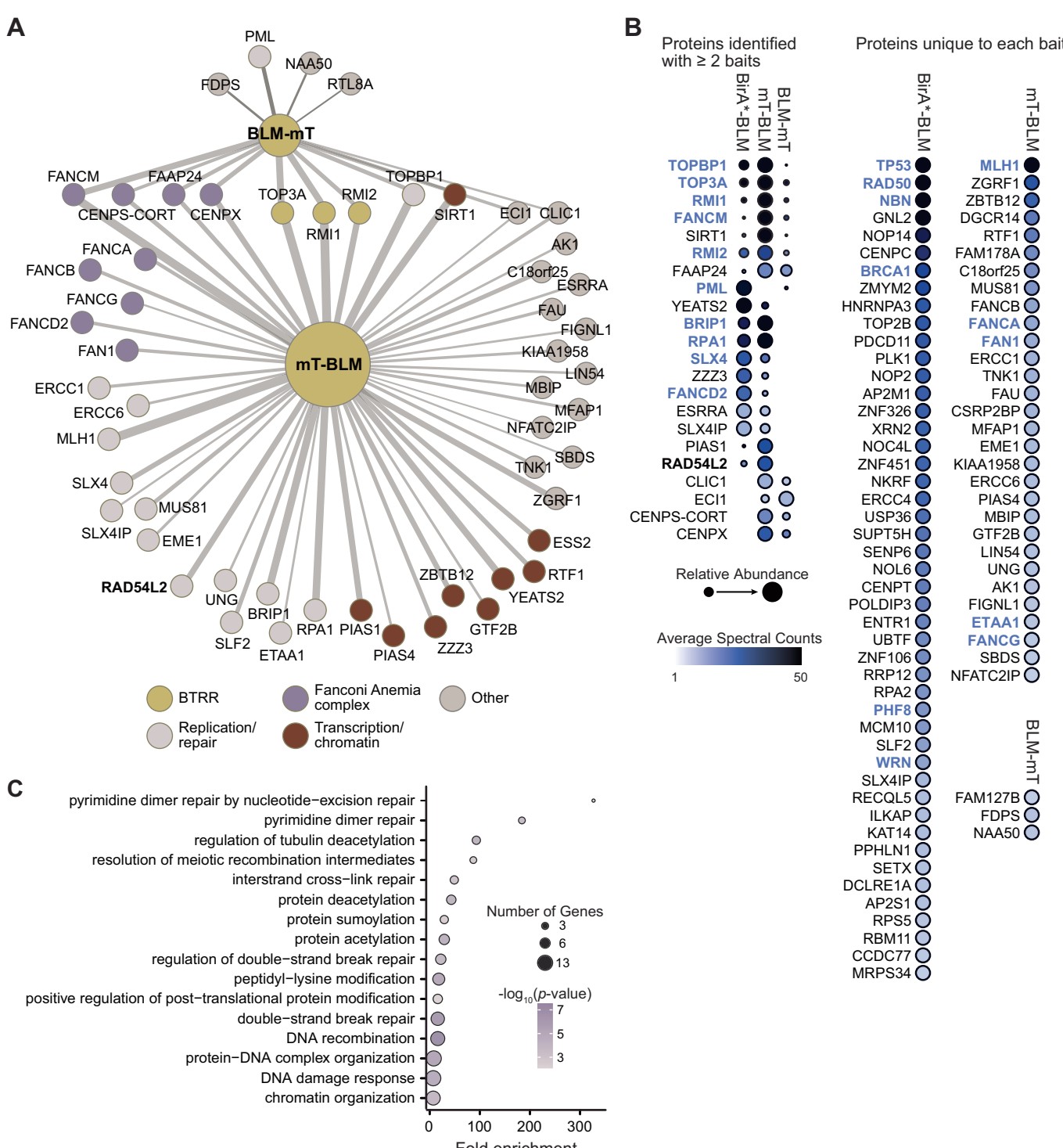

specifically present in the immunoprecipitates when BirA*FLAG-BLM was expressed (Fig. 4B). Similarly, miniTurbo-FLAG-BLM immunoprecipitates specifically contained RAD54L2 when the BLM fusion protein was expressed (Fig. 4C). We conclude that BLM/RAD54L2 complexes are stable to affinity purification, at least when BLM is overexpressed. It remains to be determined if BLM and RAD54L2 interact directly.

To gain further insight into RAD54L2 function we performed BioID with RAD54L2 (Fig. 4D and Source Data 4D). Among the most prominent high-confidence proximal interactions was BLM. There was extensive overlap with the BLM proximal proteome, including TOP3A, RMI1, TOPBP1, SLX4, RAD50, and 19 other proteins (Fig. 4D). Examining the functional enrichment of the high-confidence RAD54L2 proximal proteome (237 proteins), we

found functions in common with the BLM proximal proteome, including 'DNA damage response', 'chromosome segregation', and 'protein sumoylation' (Fig. 4E; Source Data 2C and Source Data 4E). We also noted functions consistent with roles in modulating chromatin accessibility and transcription, including "regulation of transcription", "nucleosome organization", and "transcription initiation" (Fig. 4E). Indeed, *RAD54L2* (also known as *ARIP4* (androgen receptor-interacting protein 4)) was first described as a transcriptional regulator (Rouleau et al, 2002; Sitz et al, 2004). We infer that RAD54L2 could perform distinct functions in transcription and DNA repair, and that proximity to BLM could be a major aspect of RAD54L2 function.

Figure 2. **High-resolution proximal proteome of the BLM protein.**

(A) Network of the 15 proximal proteins detected with BLM tagged with miniTurbo at the C-terminus (BLM-mT) and the 51 proximal proteins detected with BLM tagged at the N-terminus (mT-BLM). The nodes are the proximal protein preys and the edge widths are proportional to the log2 of the average spectral counts for the given bait-prey pair. Members of the Fanconi anemia complex, the BTRR complex, proteins that function in DNA replication and/or DNA repair, and proteins that function in transcription and/or chromatin function are indicated. (B) Dot plot showing prey proteins identified with BLM N- and C-terminal miniTurbo fusions. High-confidence proximal proteins are shown (BFDR ≤0.01), with the prey from the BirA*-BLM screen from Fig. 1A plotted for comparison. Each BioID screen was performed in quadruplicate, with a dot color indicating the average spectral counts. The dot outlines indicate the BFDR, and dot size indicates relative abundance across the baits. Previously described interaction partners annotated in BioGRID are shown in blue. (C) Gene ontology (GO) biological process analysis for 101 high-confidence BLM proximal proteins identified in the BirA* and miniTurbo BioID screens. The -fold enrichment for each GO term is indicated, the colors indicate the corrected *p*-values, and the size of the circles corresponds to the number of genes annotated to the given GO term. The *p*-values were calculated using the hypergeometric distribution with Bonferroni multiple hypothesis correction. Source data are available online for this figure.

## RAD54L2 promotes BLM focus formation

To develop mechanistic insight into how RAD54L2 promotes noncrossover recombination repair, we measured recruitment of BLM to sites of DNA replication stress, where BTRR function is important to restart replication forks (Chan et al, 2007; Shorrocks et al, 2021). Association of BTRR with chromatin manifests as punctate BLM foci that are infrequent in unperturbed cells but are induced in the presence of the replication stressor hydroxyurea (Fig. 5A,B). When *RAD54L2* was knocked out, BLM foci were absent, indicating that *RAD54L2* promotes the recruitment of BLM to chromatin during DNA replication stress (Fig. 5A,B). BLM functions at distinct stages of recombination repair of DSBs and stressed replication forks: BLM promotes long-range DNA resection to generate the template for RAD51-ssDNA filament formation (Gravel et al, 2008; Nimonkar et al, 2011) and it catalyzes dissolution of double Holliday junctions to suppress the formation of crossover repair products (Wu and Hickson, 2003; Wu et al, 2006; Raynard et al, 2006; Bussen et al, 2007). We tested whether RAD54L2 was promoting BLM function during the early steps of recombination repair by measuring the recruitment of the resection protein MRE11 and the RAD51 recombinase to chromatin during DNA replication stress (Fig. 5C–F). Cells that were deficient in RAD54L2 showed no changes in recruitment of the resection or recombination proteins, indicating that RAD54L2 functions downstream of the RAD51 step of recombination repair.

## RAD54L2 ATPase is required for BLM recruitment to chromatin and for crossover suppression

SNF2-family proteins like RAD54L2 often utilize a conserved ATPase domain to modulate chromatin accessibility (Dürr et al,

2006). RAD54L2 ATPase activity requires intact Walker A and Walker B motifs (Walker et al, 1982), and ATPase activity is eliminated by changing the lysine residue in the Walker A motif to alanine (Rouleau et al, 2002). We tested if the integrity of the RAD54L2 ATPase domain was important for BLM focus formation by transfecting *RAD54L2KO* cell with wildtype or Walker A mutant (K311R) *RAD54L2* and measuring BLM foci formation in the presence and absence of HU (Figs. 6A and EV5). Expression of wild-type *RAD54L2* rescued BLM focus formation to wild-type levels. By contrast, the K311R mutant did not restore the formation of BLM foci, indicating that ATPase activity of RAD54L2 is necessary for the recruitment of BLM to chromatin during replication stress.

We next tested whether RAD54L2 catalytic activity was also important for suppressing SCEs. We transfected $RAD54L2^{KO}$ cells with wild type or K311R *RAD54L2*, and measured SCEs (Figs. 6B and EV5), and found that whereas wild type *RAD54L2* reduced the number of SCEs to that seen in the parental U2OS cells, the K311R mutant had no effect on SCEs. RAD54L2 and RAD54L2$^{K311R}$ proteins expressed well, although the K311R protein level was somewhat lower than the wild type (Fig. EV5). We infer that RAD54L2 plays a catalytic role in suppressing SCEs rather than acting exclusively as a scaffold for BLM recruitment to chromatin.

As a final measure of the impact of RAD54L2 on genome stability, we asked whether RAD54L2 suppresses the formation of ultrafine anaphase bridges. Sources of ultrafine bridges include DNA replication stress and recombination intermediates (Liu et al, 2014; Chan et al, 2018), and ultrafine bridges accumulate in BLM-deficient cells (Chan et al, 2007). We detected ultrafine bridges by probing cells with antibodies to PICH (Fig. 6C,D). As expected, the knockdown of BLM by siRNA caused an increase in ultrafine bridges in anaphase cells. Knockout of RAD54L2 also caused an increase in ultrafine bridges, although to a lesser extent than siBLM (29% of anaphases versus 43% of anaphases). We conclude that RAD54L2 is important to prevent the accumulation of ultrafine anaphase bridges.

## Discussion

We present a comprehensive proximal proteome for the BLM-TOP3A-RMI1-RMI2 complex and highlight the functional importance of the SNF2-family ATPase RAD54L2 in promoting noncrossover recombination repair, likely by facilitating the association of BTRR with DNA damage sites on chromatin. Our

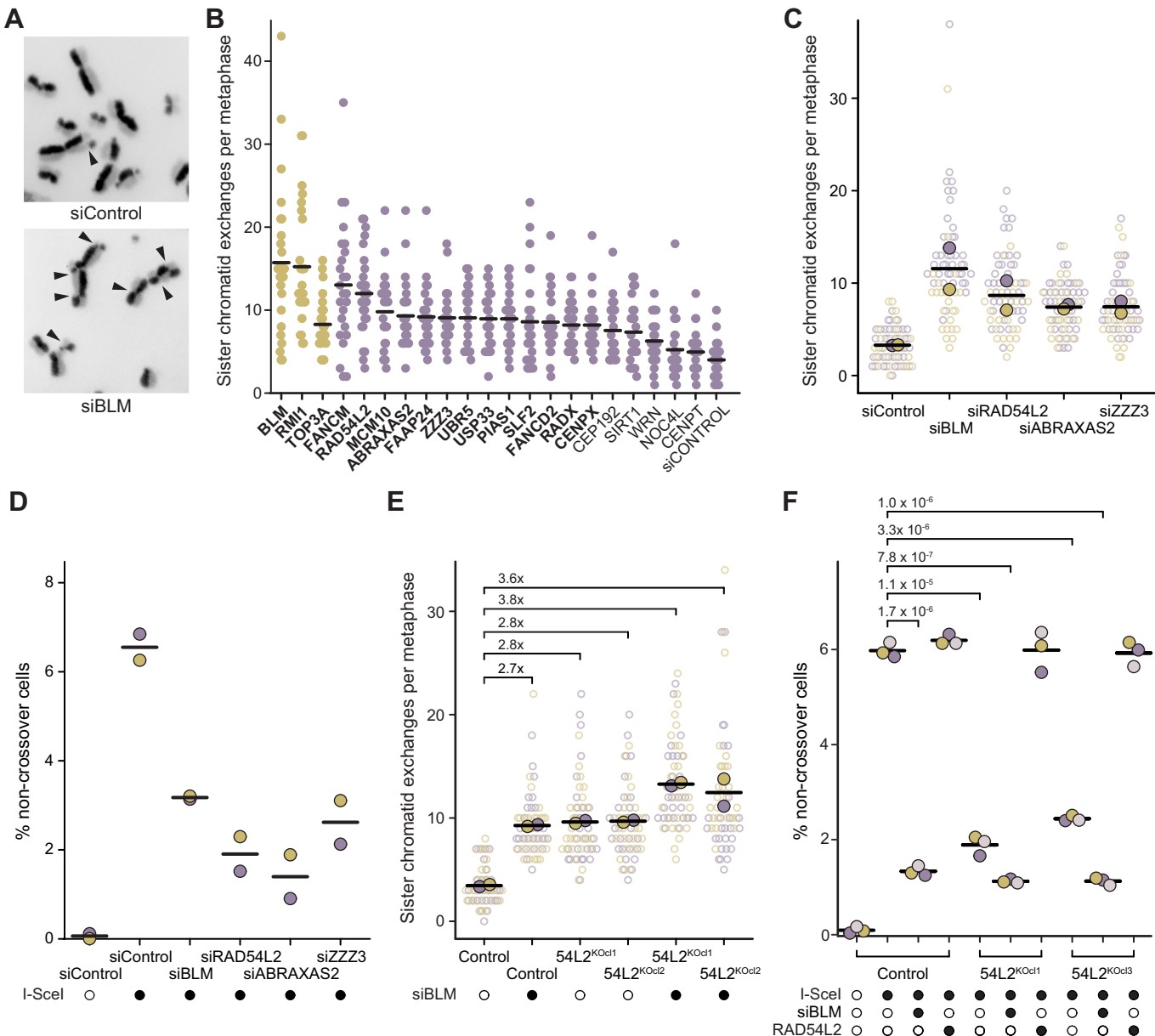

**Figure 3. BTRR proximal proteins suppress sister chromatid exchanges.**

(A) Mitotic chromosome spreads from U2OS cells following siRNA knockdown of *BLM* or control. Sister chromatids are differentially stained to detect exchanges, indicated by arrows. A nonlinear image adjustment was applied to improve the visibility of both chromatids. (B) The number of SCEs for 25 metaphases is plotted for each of the indicated gene knockdowns. Black bars indicate the means. Gene knockdowns with a >2-fold increase in SCEs are indicated in **bold**. $n = 1$. (C) The number of SCEs per metaphase is plotted for two replicates of each of the indicated gene knockdowns (25 or 40 metaphases per replicate). The replicates are indicated by the different colors and the mean SCEs for each replicate is indicated by the filled circles. The mean SCEs for each gene knockdown is indicated by the horizontal bars. $n = 2$ biological replicates. (D) The percent of cells expressing GFP (% noncrossover cells) is plotted for two replicates of each of the indicated gene knockdowns. A double-strand DNA break was induced by the expression of I-SceI where indicated (closed circles). The mean % noncrossover cells for each gene knockdown is indicated by the horizontal bars. $n = 2$ biological replicates. (E) The number of SCEs per metaphase is plotted for two replicates (27 or 28 metaphases per replicate) for each of two clonal *RAD54L2* gene disruption lines (KOcl1 and KOcl2) and for the parental cell line (Control). *BLM* was knocked down with siRNA where indicated (closed circles). The replicates are indicated by the different colors, and the mean SCEs for each replicate is indicated by the filled circles. The mean of each pair of replicates is indicated by the horizontal bars. The fold-change relative to Control is indicated for each experiment. $n = 2$ biological replicates. (F) The percent of cells expressing GFP (% noncrossover cells) is plotted for three replicates of the parental reporter line (Control) and of two clonal *RAD54L2* gene disruption lines (KOcl1 and KOcl3). A double-strand DNA break was induced by expression of I-SceI where indicated (closed circles), *BLM* was knocked down with siRNA where indicated (closed circles), and the wild-type *RAD54L2* gene was introduced into the knockout lines by transient transfection where indicated (closed circles). The replicates are indicated by the different colors, and the mean % noncrossover cells is indicated by the horizontal bars. $n = 3$ biological replicates. Source data are available online for this figure.

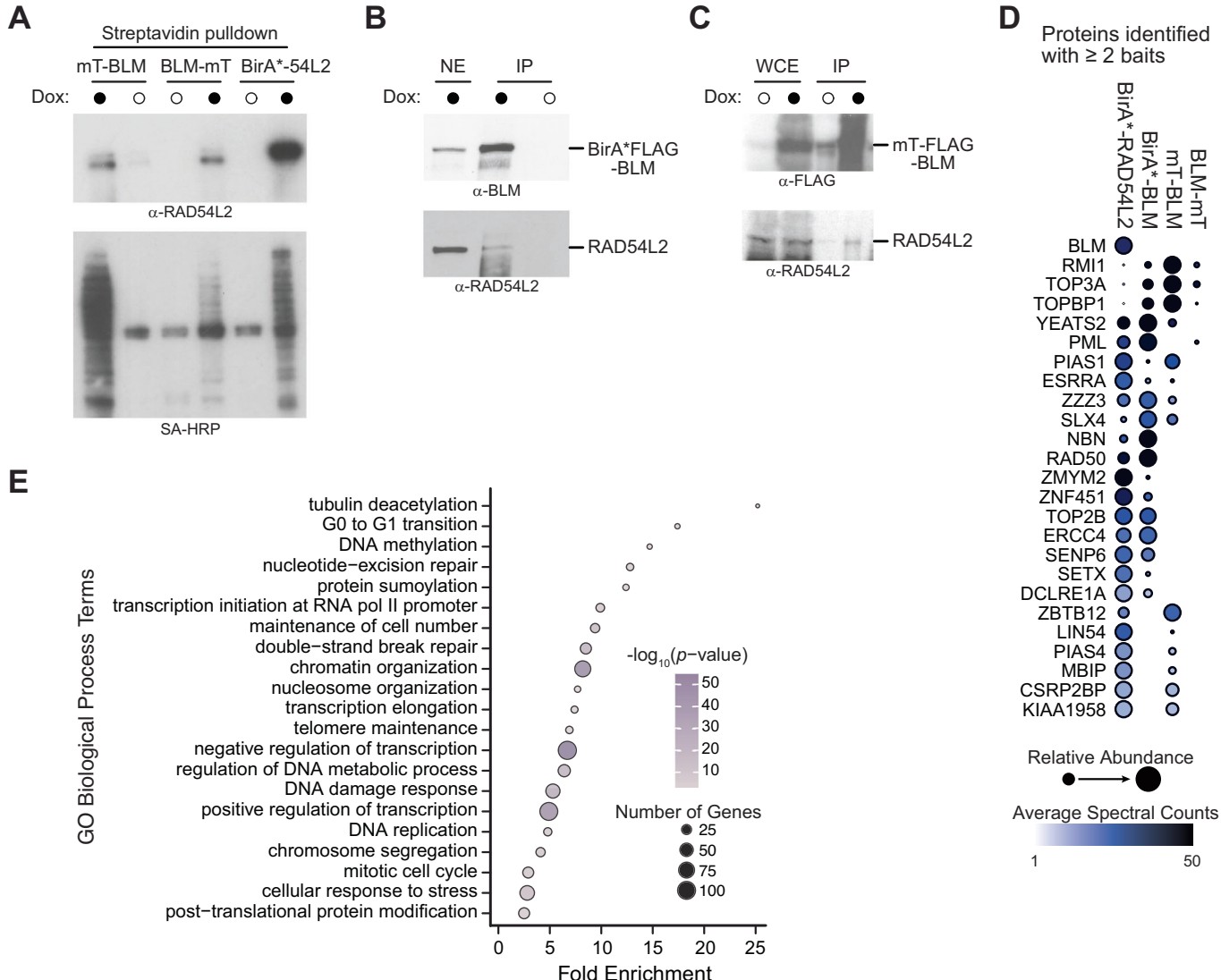

**Figure 4. The RAD54L2 proximal proteome.**

(A) Extracts of cells expressing mT-BLM, BLM-mT, or BirA*-RAD54L2, either without (open circles) or with (closed circles) doxycycline induction were affinity-purified with streptavidin-agarose. The affinity-purified biotinylated proteins were fractionated on SDS-PAGE and immunoblots were probed with anti-RAD54L2 antibodies or with streptavidin-HRP. (B) Nuclear extracts of BirA*FLAG-BLM cells, either with (closed circles) or without (open circle) doxycycline induction of BirA*FLAG-BLM expression were subjected to affinity purification with anti-FLAG-agarose. The nuclear extract (NE) and affinity-purified proteins (IP) were fractionated on SDS-PAGE and immunoblots were probed with anti-BLM or anti-RAD54L2 antibodies. The positions of BirA*FLAG-BLM and RAD54L2 are indicated. (C) Extracts of mT-FLAG-BLM cells, either without (open circles) or with (closed circles) doxycycline induction, were subjected to affinity purification with anti-FLAG-agarose to purify mT-BLM (which contains a FLAG epitope tag). The extract (WCE) and affinity-purified proteins (IP) were fractionated on SDS-PAGE and immunoblots were probed with anti-FLAG or anti-RAD54L2 antibodies. The positions of mT-FLAG-BLM and RAD54L2 are indicated. (D) Dot plot showing prey proteins identified with BirA*-RAD54L2, with the BLM BioID data from Fig. 2B plotted for comparison. High-confidence proximal proteins (BFDR ≤0.01) identified with RAD54L2 and at least one of the BLM fusions are shown. The RAD54L2 BioID screen was performed in quadruplicate, with a dot color indicating the average spectral counts. The dot size indicates relative abundance across the baits. (E) Gene ontology (GO) biological process analysis for 238 high-confidence RAD54L2 proximal proteins identified in the BirA*-RAD54L2 BioID screen. The -fold enrichment for each GO term is indicated, the colors indicate the corrected p-values, and the size of the circles corresponds to the number of RAD54L2 proximal proteins annotated to the given GO term. The p-values were calculated using the hypergeometric distribution with Bonferroni multiple hypothesis correction. Source data are available online for this figure.

data are consistent with a model (Fig. 6E) whereby RAD54L2 promotes BTRR dissolution of late recombination intermediates resulting from double-strand DNA breaks or DNA replication fork restart. RAD54L2 function appears to occur downstream of the MRE11 nuclease and the RAD51 recombinase and requires the ATPase activity of RAD54L2. The precise biochemical role of

RAD54L2 in promoting noncrossover recombination outcomes catalyzed by BTRR awaits future investigation.

Given the distinct roles of the BTRR complex at different DNA structures, including DNA replication forks (Davies et al, 2007), double Holliday junctions (Wu et al, 2006; Raynard et al, 2006), ultra-fine anaphase bridges (Chan et al, 2007), D-loops (Bachrati

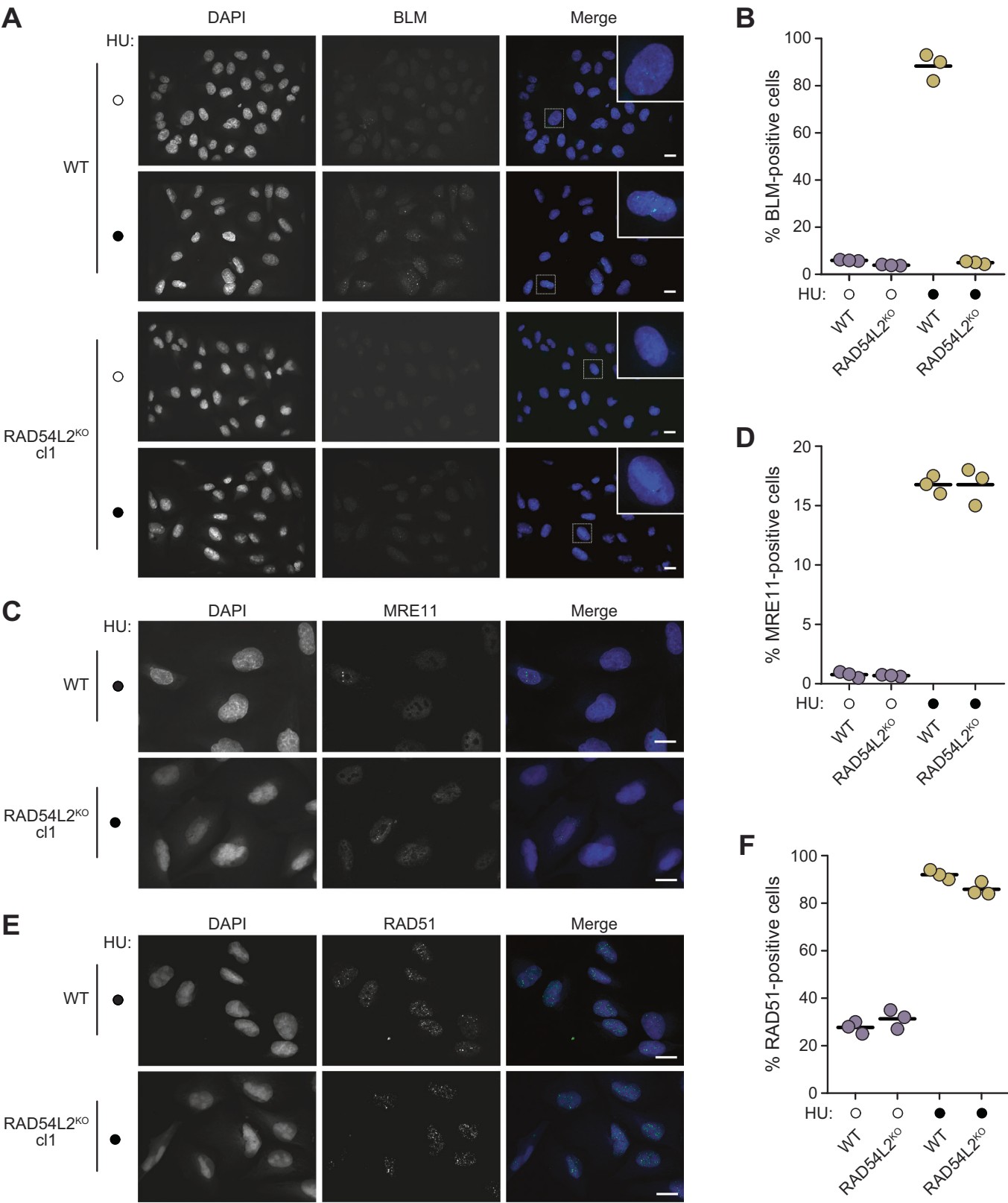

**Figure 5. Recruitment of BLM, but not MRE11 or RAD51, to chromatin requires RAD54L2.**

(A) Fluorescence micrographs of parental (WT) and *RAD54L2* knockout cells, untreated (open circles) or treated with 4 mM HU for 24 h (closed circles). Images of cells stained with DAPI to illuminate the nuclear DNA, with antibodies to BLM, and the merged images are shown. Scale bars are 20 μm. (B) The percent of parental (WT) and *RAD54L2* knockout cells displaying BLM nuclear foci following control (open circles) or HU treatment (4 mM HU for 24 h; closed circles) is plotted. Horizontal bars indicate the means. $n = 3$ biological replicates. Comparing WT + HU to *RAD54L2*$^{KO}$ + HU, $p = 1.5 \times 10^{-5}$, calculated with a two-sided Student's $t$-test. (C) Fluorescence micrographs of parental (WT) and *RAD54L2* knockout cells, treated with HU (4 mM HU for 24 h). Images of cells stained with DAPI to illuminate the nuclear DNA, with antibodies to MRE11, and the merged images are shown. Scale bars are 20 μm. (D) The percent of parental (WT) and *RAD54L2* knockout cells displaying MRE11 nuclear foci following control (open circles) or HU treatment (4 mM HU for 24 h; closed circles) is plotted. Horizontal bars indicate the means. $n = 3$ biological replicates. (E) Fluorescence micrographs of parental (WT) and *RAD54L2* knockout cells, treated with HU (4 mM HU for 24 h). Images of cells stained with DAPI to illuminate the nuclear DNA, with antibodies to RAD51, and the merged images are shown. Scale bars are 20 μm. (F) The percent of parental (WT) and *RAD54L2* knockout cells displaying RAD51 nuclear foci following control (open circles) or HU treatment (4 mM HU for 24 h; closed circles) is plotted. Horizontal bars indicate the means. $n = 3$ biological replicates. Source data are available online for this figure.

et al, 2006; Harami et al, 2022), and G-quadruplexes (Sun et al, 1998), a comprehensive interactome for the BTRR complex offers a rich dataset for identifying proteins involved in the maintenance of genome stability. Our BioID studies complement existing affinity-purification/mass spectrometry (AP-MS) datasets by capturing BTRR proximal proteins in their native, unperturbed cellular environment. AP-MS captures interactions that are stable to cell lysis conditions and are preserved or formed at the time of purification, so there is usually only a modest overlap between AP-MS and BioID-MS datasets for DNA replication and repair proteins. For example, SLX4 affinity-purification and proximity-labeling proteomes had only 7.2% overlap (Aprosoff et al, 2023), and PCNA proximal proteome showed only 11.6% overlap (Srivastava et al, 2018). Our BLM proximal proteome, from three distinct BLM BioID experiments and comprising 101 high-confidence proximity interactions, shared 25 proteins with the 183 protein BLM interactome annotated in BioGRID (Oughtred et al, 2021) (14%). Likewise, we found that our RAD54L2 high-confidence (FDR ≤0.01) proximal proteome of 237 proteins has little overlap with the 78 RAD54L2 protein interactors annotated in BioGRID (9 proteins, 11.5%) or with a recent RAD54L2 AP-MS analysis (three proteins, 1.2%) (D'Alessandro et al, 2023). These data indicate the potential for proximity-labeling methods to reveal proximal partners that are not detected using affinity purification methods.

It is, of course, important to evaluate whether a given interactome captures functional information. We find that the BTRR proximal proteome contains proteins that are known to function in concert with BTRR, including FANCM (Deans and West, 2009; Hoadley et al, 2012), additional BRAFT complex members FANCA and FANCG (Meetei et al, 2003), BRIP1/FANCJ (Suhasini et al, 2011), FANCD2 (Chaudhury et al, 2013), the RPA heterotrimer (RPA1, RPA2, and RPA3) (Yang et al, 2012; Xue et al, 2013; Doherty et al, 2005), TOPBP1 (Blackford et al, 2015), MLH1 (Langland et al, 2001; Pedrazzi et al, 2001), TP53 (Wang et al, 2001), BRCA1 (Acharya et al, 2014), RAD50 (Franchitto and Pichierri, 2002), and WRN (von Kobbe et al, 2002). As expected, the BTRR proximal proteome is strongly enriched for genes annotated to DNA replication, DNA recombination, and DNA repair gene ontology processes, and interestingly, functional enrichments are still apparent within the BTRR proximal proteome even after removing the known BTRR interactors (Fig. 1D). Additionally, we analyzed the functions of 18 proteins in the BTRR proximal proteome in suppressing sister chromatid exchanges and

identified 13 that resulted in increased SCEs when depleted (Fig. 3B). SCE suppressors include members of the FA pathway (FANCM, FAAP24, FANCD2, CENPX), a replication protein (MCM10), ubiquitylation and sumoylation regulators (ABRAXAS2 (Feng et al, 2010; Zhang et al, 2014), UBR5, USP33, PIAS1), a histone reader (ZZZ3 (Mi et al, 2018)), and genome stability proteins (RAD54L2 (D'Alessandro et al, 2023; Zhang et al, 2023), SLF2 (Räschle et al, 2015), RADX (Dungrawala et al, 2017)). The increases in SCEs that we found upon knockdown of FANCM, FAAP24, FANCD2, and SLF2 are similar to those previously reported following siRNA depletion of FANCM (Deans and West, 2009; Wang et al, 2013b), knockout of FAAP24 (Wang et al, 2013b), knockout of FANCD2 (Yamamoto et al, 2005), and in SLF2 patient fibroblasts (Grange et al, 2022). We conclude that the BTRR proximal proteome is rich in functional information.

Of the proteins in the BTRR proximal proteome, we found that RAD54L2 was biotinylated in vivo by both BirA*-BLM and miniTurbo-BLM (Figs. 1B, 2B, 4A). Both BLM fusion proteins could also immunoprecipitate RAD54L2 from nuclear or whole-cell extracts (Fig. 4B,C), suggesting that BLM and RAD54L2 are members of a stable complex. Knockdown or knockout of *RAD54L2* resulted in increased SCEs, decreased crossover recombination, and increased accumulation of ultra-fine anaphase bridges, indicating that *RAD54L2* deficiency phenocopies loss of *BLM*. Three lines of evidence suggest that *RAD54L2* is functioning in the *BLM* pathway to suppress SCEs. First, combining *BLM* knockdown with *RAD54L2* deficiency resulted in a small increase in SCEs that was neither additive nor multiplicative (Fig. 3E) and siBLM plus *RAD54L2* knockout reduced NCO recombination to the same extent as siBLM alone (Fig. 3F), indicating that *BLM* and *RAD54L2* are in the same genetic pathway. Second, there is a substantial overlap between the BLM and the RAD54L2 proximal proteomes (Fig. 4C), consistent with BLM and RAD54L2 functioning in concert. Finally, cells deficient in *RAD54L2* fail to form BLM foci on chromatin in response to DNA replication stress (Fig. 5A,B), suggesting that *RAD54L2* is important for the recruitment of BLM to late recombination intermediates. We also found that the ATPase activity of RAD54L2 is important for its ability to promote BLM focus formation (Fig. 6A) and its ability to suppress SCEs (Fig. 6B), indicating a catalytic role rather than a simple scaffolding function. Recent studies have implicated RAD54L2 in resistance to the topoisomerase poison etoposide (Zhang et al, 2023; D'Alessandro et al, 2023). RAD54L2 is proposed to function in concert with the SUMO E3 ligase ZNF451 to remove

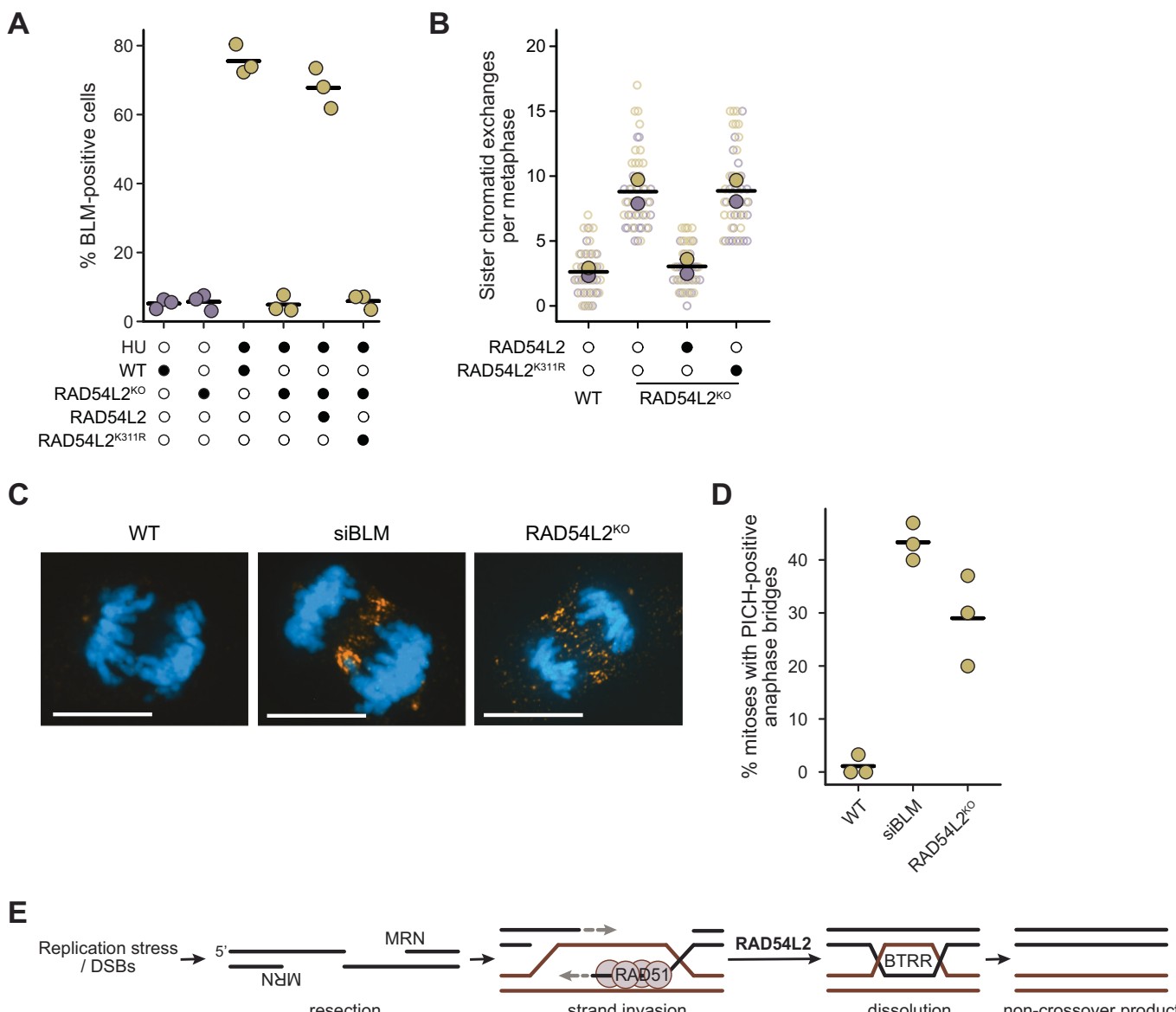

**Figure 6.  RAD54L2 ATPase activity is required to promote BLM foci and suppress SCEs.**

(A) The percent of parental (WT) and *RAD54L2* knockout cells displaying BLM nuclear foci following control (open circles) or HU treatment (4 mM HU for 24 h; closed circles) is plotted for three replicates. Where indicated by closed circles, cells were transfected with *RAD54L2* or *RAD54L2-K311R*. Horizontal bars indicate the means. $n = 3$ biological replicates. Comparing WT + HU to *RAD54L2*KO + HU, $p = 1.6 \times 10^{-5}$; WT + HU to *RAD54L2*KO + *RAD54L2* + HU, $p = 0.14$; WT + HU to *RAD54L2*KO + *RAD54L2*-*K311R* + HU, $p = 1.5 \times 10^{-5}$; all calculated with a two-sided Student's *t*-test. (B) The number of SCEs per metaphase is plotted for parental (WT) and *RAD54L2* knockout cells. Where indicated by closed circles, cells were transfected with *RAD54L2* or *RAD54L2-K311R*. The replicates are indicated by the different colors, and the mean SCEs for each replicate is indicated by the filled circles. The mean of each pair of replicates is indicated by the horizontal bars. $n = 2$ biological replicates. (C) Fluorescence micrographs of representative parental (WT), *BLM* knockdown (siBLM), and *RAD54L2* knockout (*RAD54L2*KO) anaphase cells, as quantified in panel D. Cells were stained with DAPI to illuminate the nuclear DNA and with antibodies to PICH to illuminate ultrafine anaphase bridges. Scale bars are 10 μm. (D) The percent of mitoses with ultrafine anaphase bridges is plotted for parental (WT), *BLM* knockdown (siBLM), and *RAD54L2* knockout (*RAD54L2*KO) cells. Horizontal bars indicate the means. $n = 3$ biological replicates. Comparing WT to siBLM, $p = 4.5 \times 10^{-5}$; WT to *RAD54L2*KO, $p = 5.0 \times 10^{-3}$; both calculated with a two-sided Student's *t*-test. (E) Model of the role of RAD54L2 in dissolution of DNA repair intermediates by BTRR. See text for details. Source data are available online for this figure.

trapped TOP2 cleavage complexes from DNA. Interestingly, ZNF451 was among the high-confidence proximity interactions in our RAD54L2 BioID data (Fig. 4C) and in our BLM and RMI1 BioID screens (Fig. 1A). It remains to be seen whether ZNF451 participates in RAD54L2-mediated suppression of SCEs. RAD54L2

was associated with poor clinical outcomes in gastrointestinal stromal tumors (Schoppmann et al, 2013) and in a pediatric AML cohort treated with etoposide (Nguyen et al, 2023). Given that RAD54L2 is a potential prognostic biomarker and drug target, the RAD54L2 and BLM proximal proteomes could have clinical utility.

# Methods

### Reagents and tools table

| Reagent/resource | Reference or source | Identifier or catalog number |
| --- | --- | --- |
| **Experimental Models** | | |
| HEK293 cells (*H. sapiens*) | ATCC | CRL-1573 |
| U-2 OS cells (*H. sapiens*) | ATCC | HTB-96 |
| **Recombinant DNA** | | |
| pCR4-TOPO-BLM | MGC Project Team et al, (2009) | N/A |
| pOTB7-RMI2 | MGC Project Team et al, (2009) | N/A |
| pcDNA5/FRT/TO-Flag-BirA* | Lambert et al, (2015) | N/A |
| pcDNA5/FRT/TO-Flag-TOP3A | Yang et al, (2012) | N/A |
| pcDNA5/FRT/TO-Flag-RMI1 | Yang et al, (2012) | N/A |
| pENTR223.1-RAD54L2 | MGC Project Team et al, (2009) | N/A |
| pcDNA3/BLM | Gaymes et al, (2002) | N/A |
| pDEST-miniTurbo-Nterm | Branon et al, (2018) | N/A |
| pDEST-miniTurbo-Cterm | Branon et al, (2018) | N/A |
| pCBAScel | Richardson et al, (1998) | N/A |
| PX459 | Ran et al, (2013) | N/A |
| **Antibodies** | | |
| Mouse anti-FLAG M2 | Sigma-Aldrich | F3165 |
| Mouse anti-$\alpha$-tubulin | Cell Signaling Technologies | 3873 |
| Mouse anti-GAPDH | Cell Signaling Technologies | 97166 |
| Mouse anti-V5 | Cell Signaling Technologies | D3H8Q |
| Rabbit anti-BLM | Cell Signaling Technologies | 2742 |
| Goat anti-BLM | Bethyl Laboratories | A300-120A |
| Rabbit anti-ABRAXAS2 | Abcam | ab68801 |
| Rabbit anti-RAD54L2 | ABclonal | A6144 |
| Rabbit anti-ZZZ3 | Bethyl Laboratories | A303-331A |
| Rabbit anti-TOP3A | Proteintech | 14525-1-AP |
| Rabbit anti-RMI1 | Raise in house | 6534 |
| Rabbit anti-RAD51 | Abcam | ab133534 |
| Mouse anti-MRE11 | Novus Biologicals | NB100-473 |
| Mouse anti-PICH | Sigma-Aldrich | 04-1540 |
| Goat anti-mouse IgG HRP | Invitrogen | 31430 |
| Goat anti-rabbit IgG HRP | Invitrogen | 31460 |
| Donkey anti-Goat IgG, Alexa Fluor 546 | Invitrogen | A-11056 |
| Donkey anti-Rabbit IgG, Alexa Fluor 488 | Invitrogen | A-21206 |
| Donkey anti-Mouse IgG, Alexa Fluor 488 | Invitrogen | A-21202 |

| Reagent/resource | Reference or source | Identifier or catalog number |
| --- | --- | --- |
| Streptavidin-HRP | Cell Signaling Technologies | 3999 |
| **Oligonucleotides and other sequence-based reagents** | | |
| PCR primers | This study | N/A |
| siRNAs | Dharmacon/Horizon | N/A |
| **Chemicals, Enzymes and other reagents** | | |
| DMEM | Wisent | 319-005-CL |
| McCoy's 5 A medium | Gibco | 16600082 |
| Fetal bovine serum (FBS) | Wisent | 098-150 |
| Penicillin/streptomycin | Wisent | 450-201-EL |
| Blasticidin | Bioshop | BLA477.25 |
| Zeocin | InvivoGen | ant-zn-05 |
| Hygromycin | BioShop | HYG002.202 |
| MycoAlert PLUS Mycoplasma Detection Kit | Lonza | LT07-705 |
| Lipofectamine 3000 | Thermo Fisher | L3000001 |
| Lipofectamine RNAiMAX | Thermo Fisher | 13778075 |
| X-tremeGENE HP | Roche | 6366236001 |
| Doxycycline | Sigma-Aldrich | D9891 |
| Biotin | Sigma-Aldrich | B-4639 |
| Streptavidin-sepharose bead | GE | 17-5113-01 |
| Bromodeoxyuridine | Sigma-Aldrich | B5002 |
| Colcemid | Gibco | 15212012 |
| cOmplete EDTA-free Protease Inhibitor Cocktail | Roche | 11873580001 |
| PhosSTOP phosphatase inhibitor | Roche | 4906845001 |
| Pierce Rapid Gold BCA Protein Assay Kit | Thermo Fisher | A53225 |
| SuperSignal West Pico PLUS chemiluminescent substrate | Thermo Fisher | 34580 |
| Dynabeads Protein G | Invitrogen | 10003D |
| Hydroxyurea | BioBasic | HB0528 |
| Goat serum | Thermo Fisher | 16210064 |
| 4',6-diamidino-2-phenylindole (DAPI) | Sigma-Aldrich | D9542 |
| ProLong Gold Antifade | Thermo Fisher | P36930 |
| **Software** | | |
| ProHits 7.0 Laboratory Information Management System | Liu et al, (2016) | N/A |
| ProteoWizard | https://proteowizard.sourceforge.io/ | N/A |
| Mascot | Perkins et al, (1999) | N/A |
| Comet | Eng et al, (2013) | N/A |
| R | https://www.r-project.org/ | N/A |
| FlowJo | https://www.flowjo.com/ | N/A |

| Reagent/resource | Reference or source | Identifier or catalog number |
|---|---|---|
| **Other** | | |
| LTQ-Orbitrap Velos | Thermo Electron | N/A |
| Orbitrap Elite | Thermo Electron | N/A |
| AxioImager Z1 | Zeiss | N/A |
| AxioObserver Z1 | Zeiss | N/A |
| LSR II | BD Biosciences | N/A |

## Plasmids

To construct plasmids for generating stable BirA* cell lines for BioID, *BLM* and *RMI2* cDNAs were amplified from pCR4-TOPO-BLM and pOTB7-RMI2 (MGC Project Team et al, 2009). The PCR products were digested with AscI and XhoI and cloned into pcDNA5/FRT/TO-Flag-BirA* (Lambert et al, 2015) to yield pcDNA5/FRT/TO-FLAG-BirA*-BLM and pcDNA5/FRT/TO-FLAG-BirA*-RMI2. pcDNA5/FRT/TO-Flag-TOP3A and pcDNA5/FRT/TO-Flag-RMI1 (Yang et al, 2012) were digested with AscI and XhoI and cloned into pcDNA5/FRT/TO-Flag-BirA* to yield pcDNA5/FRT/TO-FLAG-BirA*-TOP3A and pcDNA5/FRT/TO-FLAG-BirA*-RMI1. *RAD54L2* cDNA from Mammalian Gene Collection (MGC Project Team et al, 2009) was cloned into pcDNA5/FRT/TO-Flag-BirA* to yield pcDNA5/FRT/TO-FLAG-BirA*-RAD54L2.

Plasmids for miniTurbo (Branon et al, 2018) were constructed with BLM from pcDNA3/BLM (Gaymes et al, 2002), cloned into pDONR201 to yield pDONR201-BLM. The BLM cDNA was moved from pDONR201-BLM into pDEST-miniTurbo-Nterm and pDEST-miniTurbo-Cterm (Branon et al, 2018) obtained from the Gingras Lab, using a Gateway LR reaction to yield pDEST-miniTurbo-Nterm-BLM and pDEST-miniTurbo-Cterm-BLM.

For rescue experiments, the *RAD54L2* cDNA was assembled from a partial cDNA clone encoding Met109 to Lys1467 was from the human ORFeome collection (Lamesch et al, 2007) and a synthetic DNA fragment encoding Met1 to Glu108 (Invitrogen). *RAD54L2*$^{K311R}$ was constructed by QuikChange site-directed mutagenesis with primers (5′-AGATCACTTGCAAAGTTCTCCCCAGACCCATGCTG-3′ and 5′-CAGCATGGGTCTGGGGAGAACTTTGCAAGTGATCT-3′). Both constructs were confirmed by whole plasmid sequencing.

To generate RAD54L2 knockouts a double-stranded DNA oligo encoding a sgRNA targeting the RAD54L2 5′ region from the TKOv3 library (Hart et al, 2017) (5′-AAGATGGGCAGCAGCCGCCGCGG–3′) was cloned into pSpCas9(BB)-2A-Puro (PX459) (RRID: Addgene_48139) to yield PX459-RAD54L2.

All constructs were confirmed by Sanger sequencing unless otherwise noted.

## Human cell culture

Flp-In-T-REx-293 (RRID: CVCL_U427) cells were grown in Dulbecco's modified Eagle's medium (DMEM; Wisent #319-005-CL) supplemented with 10% fetal bovine serum (FBS; Wisent #098-150), 1% penicillin/streptomycin (Wisent #450-201-EL), 15 µg/mL

blasticidin (BioShop #BLA477.25), and 100 µg/mL zeocin (Invivo-Gen #ant-zn-05). Stably transfected Flp-In-T-REx-293 lines were cultured with 15 µg/mL blasticidin (BioShop #BLA477.25) and 200 µg/mL hygromycin (BioShop # HYG002.202). U2OS (RRID: CVCL_0042) and U2OS DR-GFP (RRID:CVCL_B0A7) cells were grown in McCoy's 5 A medium (Gibco # 16600082) supplemented with 10% fetal bovine serum (FBS; Wisent #098-150) and 1% penicillin/streptomycin (Wisent #450-201-EL). All cell lines were grown at 37 °C and 5% $CO_2$. Cell lines were routinely monitored to ensure the absence of *Mycoplasma* contamination using the MycoAlert PLUS Mycoplasma Detection Kit (Lonza #LT07-705).

## Production of stable cell lines for BioID and miniTurbo proximity-dependent biotinylation

To generate stable cell lines, Flp-In-T-REx-293 cells were transfected with the BioID or miniTurbo construct of interest plus the Flp recombinase expression vector pOG44, using Lipofectamine 3000 (Thermo Fisher # L3000001) or X-tremeGENE HP (Roche # 6366236001), according to the manufacturer's instructions. At 24 h post-transfection, cells were expanded for selection with 200 µg/mL hygromycin B. Hygromycin-resistant cells were expanded in the presence of 200 µg/mL hygromycin to create polyclonal stable cell lines. Where indicated, expression of fusion proteins was induced for 24 h by the addition of doxycycline (Sigma-Aldrich # D9891) to 5 µg/ml. The expression of BirA* and miniTurbo fusion proteins was confirmed by immunoblotting (Expanded View Figs. EV1, EV2).

## Streptavidin purification of biotinylated proteins and on-bead trypsin digest

Proximal proteins of BLM, TOP3A, RMI1, and RMI2 were identified as previously described (Lambert et al, 2015). Proximal proteins in RAD54L2 BioID and BLM miniTurbo were prepared as previously described (St-Germain et al, 2020). For each biological replicate, HEK293 cells stably expressing a tetracycline regulated BirA* (two 150 mm dishes) or miniTurbo fusion protein (five 150 mm dishes) were grown to 60-80% confluency. Cells were treated with 1 µg/mL doxycycline and 50 µM biotin (Sigma-Aldrich # B-4639) for 24 h for BirA* fusions or 30 min for miniTurbo fusions. Medium was removed and cells were washed twice with cold PBS. Cells were scraped and pooled together and washed two times with cold PBS. Cell pellets were flash-frozen and stored at −80 °C.

The BirA* expressing cells from two 150 mm dishes were lysed at 1:10 cell pellet weight (g) : RIPA buffer volume (mL) (50 mM Tris-HCl pH 7.5, 150 mM NaCl, 1% Triton X-100, 1 mM EDTA, 1 mM EGTA, 0.1% SDS, Sigma protease inhibitors P8340 1:500, and 0.5% sodium deoxycholate), for 1 h on a nutator at 4 °C. For miniTurbo fusions, five 150 mm dishes were incubated at 4 °C in 10 mL RIPA buffer, for 1 h on a nutator. After incubation, 1 µl of benzonase (250U) was added to each sample, and the lysates were sonicated (3 × 10 s bursts with 2 s rest in between) on ice at 65% amplitude. The lysates were then centrifuged for 30 min at 20,817 × *g* at 4 °C. During this step, streptavidin-sepharose beads (GE # 17-5113-01) were washed three times with 1 mL RIPA buffer (minus protease inhibitors and sodium deoxycholate). Beads were pelleted at 400×*g* for 1 min in between washes. After centrifugation, supernatants from the clarified lysates were transferred to 15 mL

falcon tubes, and a 30 μL bed volume of washed beads was added to each. Affinity purification was performed at 4 °C on a nutator for 3 h, then beads were pelleted (400×g, 1 min), the supernatant removed, and the beads transferred to a 1.5 mL Eppendorf tube in 1 mL RIPA buffer (minus protease inhibitors and sodium deoxycholate). The beads were washed by pipetting up and down (four times per wash step) with 1 mL RIPA buffer (minus protease inhibitors and sodium deoxycholate), followed by two washes in TAP lysis buffer (50 mM HEPES-KOH pH 8.0, 100 mM KCl, 10% glycerol, 2 mM EDTA, 0.1% NP-40), then three washes in 50 mM ammonium bicarbonate pH 8 (ABC). Beads were pelleted by centrifugation (400×g, 1 min), and the supernatant was aspirated between wash steps. After the last wash, all residual 50 mM ABC was pipetted off.

Beads were resuspended in 30 μL (for BioID) and 200 μL (for miniTurbo) of 50 mM ammonium bicarbonate (ABC) pH 8 with 1 μg trypsin added and incubated at 37 °C overnight while mixing on a rotating disc. The next day, an additional 0.5 μg of trypsin was added to each sample (in 10 μL 50 mM ABC), and the samples were incubated for an additional 2 h at 37 °C with mixing on a nutator. Beads were pelleted (400×g, 2 min), and the supernatant was transferred to a fresh 1.5 mL Eppendorf tube. The beads were then rinsed two times with 30 μL of HPLC $H_2O$ for BirA* fusions or 150 μl of 50 mM ABC for miniTurbo fusions each time (pelleting beads at 400×g, 2 min in between), and these rinses combined with the original supernatant. The pooled supernatant was acidified by adding 50% formic acid (FA) to a final concentration of 2% v/v and dried by vacuum centrifugation. For BirA* fusions, dried samples were resuspended in 12 μL of 5% FA.

## Mass spectrometric analysis

Affinity-purified digested material in 12 μL of 5% formic acid was centrifuged at 16,100×g for 1 min before 6 μL was loaded onto silica columns pre-packed with 10–12 cm of C18 reversed-phase material (ZorbaxSB, 3.5 μm). The loaded column was placed in-line with an LTQ-Orbitrap Velos or an Orbitrap Elite (Thermo Electron, Bremen, Germany) equipped with a nanoelectrospray ion source (Proxeon Biosystems, Odense, Denmark) connected in-line to a NanoLC-Ultra 2D plus HPLC system (Eksigent, Dublin, USA). The LTQ-Orbitrap Velos or Orbitrap Elite instrument under Xcalibur 2.0 was operated in the data-dependent mode to automatically switch between MS and up to 10 subsequent MS/MS acquisitions. Buffer A is 100 parts $H_2O$, 0.1 part formic acid; buffer B is 100 parts acetonitrile, 0.1 part formic acid. The HPLC gradient program delivered an acetonitrile gradient over 125 min. For the first twenty minutes, the flow rate was 400 μL/min at 2%B. The flow rate was then reduced to 200 μL/min, and the fraction of solvent B increased in a linear fashion to 35% at 95.5 min. Solvent B was then increased to 80% over 5 min and maintained at that level until 107 min. The mobile phase was then reduced to 2% B until the end of the run (125 min). The parameters for data-dependent acquisition on the mass spectrometer were 1 centroid MS (mass range 400–2000) followed by MS/MS on the ten most abundant ions. General parameters were: activation type = CID, isolation width = 1 m/z, normalized collision energy = 35, activation Q = 0.25, activation time = 10 ms. For data-dependent acquisition, the minimum threshold was 500, the repeat count = 1, the repeat duration = 30 s, the exclusion size list = 500, exclusion duration = 30 s, and exclusion mass width (by mass) = low 0.03, high 0.03.

## Mass spectrometry data analysis

All mass spectrometry data files were stored, searched, and analyzed using the ProHits 7.0 Laboratory Information Management System (LIMS) platform (Liu et al, 2016). Within ProHits, RAW files were converted to an MGF format using ProteoWizard (V3.0.10702). The data were searched using Mascot V2.3.02 (Perkins et al, 1999) and Comet V2018.01 rev.4 (Eng et al, 2013). The spectra were searched with the human and adenovirus sequences in the NCBI Reference Sequence Database (version 57, January 30, 2013), supplemented with "common contaminants" from the Max Planck Institute (http://maxquant.org/contaminants.zip) and the Global Proteome Machine (GPM; ftp://ftp.thegpm.org/fasta/cRAP/crap.fasta), forward and reverse sequences (labeled "gi|9999" or "DECOY"), sequence tags, and streptavidin, for a total of 72,482 entries. Database parameters were set to search for tryptic cleavages, allowing up to 2 missed cleavages per peptide with a mass tolerance of 12 ppm for precursors with charges of 2+ to 4+ and a tolerance of 0.6 Da for fragment ions. Variable modifications were selected for deamidated asparagine and glutamine, and oxidated methionine. Results from each search engine were analyzed through TPP (the Trans-Proteomic Pipeline, v.4.7 POLAR VORTEX rev 1) via the iProphet pipeline (Shteynberg et al, 2011). All proteins with an iProphet probability ≥95% were used for analysis.

## Protein proximity interaction scoring

Significance analysis of interactome express (SAINTexpress) version 3.6.3 was used to calculate the probability of potential associations/interactions compared to background contaminants using default parameters (Teo et al, 2014). In brief, SAINTexpress is a statistical tool that compares the spectral counts of each prey identified with a given biotin ligase-tagged bait against a set of negative controls. For BioID, negative controls consisted of streptavidin affinity purifications from cells expressing BirA*-FLAG and BirA*, 6 biological replicates each. Two or more biological replicates were collected for all cell lines and conditions. For BioID proximity interaction scoring, two replicates with the highest spectral counts for each prey were used for baits; four replicates were used for negative controls for SAINTexpress. SAINT scores were averaged across replicates, and these averages were used to calculate a Bayesian false discovery rate (BFDR); preys with BFDR ≤1% were considered high-confidence protein interactions. All non-human protein interactors (did not start with "NP" in the prey column) were removed from the SAINT analysis. Proximity interaction scores are tabulated in SourceData1AB.xlsx, SourceData2B.xlsx, and SourceData4C.xlsx.

## Gene ontology term enrichment analysis

GO enrichment analysis was performed using high-confidence proximity interactors as inputs for the Princeton Generic Gene Ontology (GO) Term Finder (Boyle et al, 2004). Redundant GO terms were consolidated with REVIGO (Supek et al, 2011) using a medium list size and the SimRel semantic similarity measure, and in some cases manually, and terms that included more than 10% of genes were removed. The complete unfiltered outputs from GOTermFinder are provided in SourceData1D.xlsx, SourceData2C.xlsx, and SourceData4D.xlsx.

## Protein–protein interaction data mining

Physical interactions with low-throughput or high-throughput evidence for BTRR complex members were downloaded from BioGRID (https://thebiogrid.org/; accessed 24/09/2023; SourceData1AB.xlsx). The physical interaction network for BTRR BioID hits (Fig. 1C) was generated with GeneMANIA (https://genemania.org/; accessed 23/01/2024; SourceData1C.xlsx). The BLM proximity interaction network was generated from the miniTurbo data using Cytoscape 3.10.0 (Shannon et al, 2003).

## Gene knockdowns with RNA interference

For the assays in Fig. 3, U2OS cells were transfected with small interfering RNA (siRNA) pools (Dharmacon/Horizon) or individual siRNAs, as listed in the Reagents and Tools Table. An siRNA that does not have a target in the human genome was used as the negative control. All experiments were performed 48 h after siRNA transfection to achieve optimal protein depletion, unless indicated otherwise. Lipofectamine RNAiMAX (Invitrogen) was used to carry out siRNA transfections.

## Sister chromatid exchange assays

Approximately $0.2 \times 10^6$ U2OS cells per well were seeded and grown overnight. U2OS cells were then cultured for 48 h (approximately two rounds of DNA replication) in a medium supplemented with 10 μM bromodeoxyuridine (BrdU; Sigma-Aldrich # B5002) to differentially label sister chromatids. Cells were treated with 100 ng/μl colcemid (Gibco # 15212012) for 2 h to arrest the cells in metaphase. The cells were harvested, resuspended in 10 mL 0.075 M KCl, and incubated at 37 °C for 20 min. The swollen cells were pelleted by centrifugation, the supernatant was removed, and the cells were resuspended with 10 mL of fresh 3:1 methanol:acetic acid fixative solution. The cells were washed with the fixative solution twice. After the last fixative wash, the cells were stored in 1 mL of the fixative solution. Microscope slides were cleaned with ethanol, dried, and steamed for a few seconds above a beaker of boiling water immediately before chromosome spreading. The cell suspension was dropped onto the steamed slide until the surface was covered, slides were air dried, and quickly dipped in 70% acetic acid. Slides were then cured in the dark for 24 h at room temperature. Cells were stained with 0.75 μg/ml Hoechst 3342 for 30 min and exposed to UV light for 30 min, washed with 2X SSC buffer for 15 min, and stained with 0.1 μg/ml DAPI in PBS. The resulting metaphase chromosome spreads were imaged on a Zeiss AxioImager Z1 widefield fluorescence microscope equipped with DAPI filter cube 49. A minimum of 25 mitoses were analyzed per experiment. Statistical support was evaluated with the Student's $t$-test.

## Direct-repeat GFP recombination assays

U2OS cells carrying the DR-GFP reporter (Pierce et al, 1999) were transfected with siRNAs as indicated above. After 24 h, cells were transfected with the I-SceI expression plasmid pCBASceI (Richardson et al, 1998) using X-tremeGENE HP. 48 h after I-SceI transfection, cells were harvested and resuspended in PBS buffer ($Ca^{2+}/Mg^{2+}$ Free) plus 0.5% heat-inactivated FBS. Live cells were harvested in FACS buffer ($Ca^{2+}/Mg^{2+}$ -free PBS with 0.5% heat-inactivated FBS) and incubated on ice during data acquisition. Flow cytometry to identify GFP-positive cells was performed on BD LSR II Flow Cytometer, and 10,000 events were acquired based on the first FSC/SSC gate. Flow cytometry data were analyzed using FlowJo (BD Biosciences, RRID: SCR_008520). For rescue experiments with RAD54L2 expression, cells were transfected with PX459-RAD54L2 and PX459-K311R and grown for 24 h before proceeding with pCBASceI transfection. Statistical support was evaluated with the Student's $t$-test.

## Construction of RAD54L2 knockout U2OS lines by CRISPR-CAS editing

U2OS or U2OS DR-GFP cells were transfected with PX459-sgRAD54L2 using X-tremeGENE HP and selected with 1.5 μg/ml puromycin for 72 h. Individual cells were isolated by flow cytometry, expanded, and screened for RAD54L2 expression by western blotting (Expanded View Fig. EV4A,C). Editing of the RAD54L2 genomic locus was confirmed by amplifying ~250 bp upstream and downstream of the Cas9 cut site from genomic DNA followed by Sanger sequencing. The Cas9 edits in RAD54L2 alleles were inferred from the Sanger sequencing data using ICE (Conant et al, 2022):

| Summary Table: Cas9 edits in RAD54L2 alleles | | |
|---|---|---|
| **RAD54L2 KO** | **KO Score** | **Indels** |
| U2OS cl.1 | 98 | {'-14': 98.0} |
| U2OS cl.2 | 71 | {'7': 2.0, '-1': 52.0, '-4': 3.0, '-7': 4.0, '-8': 1.0, '10': 2.0, '12': 1.0, '19': 3.0, '-11': 1.0, '-17': 1.0, '-28': 2.0} |
| DR-GFP cl.3 | 97 | {'1': 65.0, '-1': 32.0} |
| DR-GFP cl.1 | 96 | {'1': 31.0, '2': 33.0, '-10': 32.0} |

## Antibodies

Primary antibodies used in this study were: mouse anti-FLAG M2 (Sigma-Aldrich #F3165, RRID: AB_259529, 1:1000-1:2500), mouse anti-$\alpha$-tubulin (Cell Signaling Technologies #3873, RRID: AB_1904178, 1:5000), mouse anti-GAPDH (Cell Signaling Technologies #97166, RRID: AB_2756824, 1:5000), mouse anti-V5 (Cell Signaling Technologies #D3H8Q; 1:1000), anti-BLM (Cell Signaling Technologies #2742; Bethyl Laboratories #A300-120A; 1:1000 for immunofluorescence), anti-ABRAXAS2 (Abcam #ab68801; 1:1000), anti-RAD54L2 (ABclonal #A6144, 1:1000), anti-ZZZ3 (Bethyl Laboratories #A303-331A; 1:1000), anti-TOP3A (Proteintech # 14525-1-AP; 1:1000), anti-RMI1 (Raised in house #6534, 1:500), anti-RAD51 (Abcam #ab133534), anti-MRE11(Novus Biologicals #NB100-473, 1:1000), anti-PICH (Sigma-Aldrich #04-1540). Secondary antibodies used for immunoblotting were: goat anti-mouse IgG HRP (Invitrogen # 31430, 1:5000) and goat anti-rabbit IgG HRP (Invitrogen # 31460, 1:5000). Secondary antibody used for immunofluorescence were Donkey anti-Goat IgG, Alexa Fluor 546 (Invitrogen # A-11056, 1:1000) and Donkey anti-Rabbit

IgG, Alexa Fluor 488 (Invitrogen # A-21206, 1:1000), and Donkey anti-Mouse IgG, Alexa Fluor 488 (Invitrogen # A-21202, 1:1000).

Streptavidin-HRP (Cell Signaling Technologies #3999; 1:5000) was used to detect biotinylated proteins.

## Immunoblotting and immunoprecipitation

Whole-cell extracts for immunoblotting were prepared by lysing $1–2 \times 10^6$ cells in RIPA buffer (50 mM Tris pH 7.5, 150 mM NaCl, 0.1% SDS, 2 mM EDTA, 0.5% sodium deoxycholate, 1% (v/v) Triton X-100) containing 1x cOmplete EDTA-free Protease Inhibitor Cocktail (Roche) and 1x PhosSTOP phosphatase inhibitor (Roche) for 30 min at 4 °C with gentle agitation. Extracts were clarified by centrifugation at 14,000 rpm for 15 min at 4 °C, and the supernatant was aliquoted into new 1.5 mL tubes. Protein concentrations were measured using the Pierce Rapid Gold BCA Protein Assay Kit (Thermo Fisher #A53225). Whole-cell extracts (20–50 µg) were diluted in 4x Laemmli sample buffer (250 mM Tris pH 6.8, 5% SDS, 40% (w/v) glycerol, 0.02% bromophenol blue, 10% $\beta$-mercaptoethanol) and boiled at 95 °C for 5 min prior to fractionation by SDS-PAGE. Proteins were transferred to nitrocellulose membranes, blocked for 1 h in 5% (w/v) skim milk in TBS with 0.2% Tween-20 (TBST) and probed with primary antibodies overnight at 4 °C. Membranes were washed three times in TBST for 5 min each, probed with secondary antibodies for 1 h, then washed another three times in TBST for five minutes each. Proteins were detected using SuperSignal West Pico PLUS chemiluminescent substrate (Thermo Fisher #34580) and visualized on X-ray film.

Nuclear extracts for co-immunoprecipitations of BirA*-BLM and RAD54L2 were prepared by resuspending cell pellets in cytoplasmic lysis buffer (50 mM Tris pH 7.5, 10 mM NaCl, 1.5 mM MgCl$_2$, 10% glycerol, 0.34 M sucrose, 0.1% Triton X-100), harvesting nuclei by centrifugation at 14,000 rpm for 10 min at 4 °C, followed by resuspending in nuclear extraction buffer (50 mM Tris pH 7.5, 250 mM NaCl, 10% glycerol, 1 mM EDTA). The nuclear extract was clarified by centrifugation at 14,000 rpm for 10 min at 4°, and the supernatant (nuclear extract) was transferred to a fresh tube.

For co-immunoprecipitations, whole cell or nuclear extracts were incubated with 2.5 µg of anti-FLAG antibody bound to 50 µl Dynabeads Protein G (Invitrogen #10003D) overnight at 4 °C with gentle rotation. Immunoprecipitates were washed three times with cytoplasmic lysis buffer and eluted in 2X Laemmli buffer (4% SDS, 120 mM Tris-HCl, 0.005% bromophenol blue, 20% glycerol, 200 mM DTT) at 95 °C for 10 min. S. Proteins were fractionated by SDS-PAGE and analyzed by immunoblotting as described above.

## Immunofluorescence microscopy

For the detection of BLM, MRE11, and RAD51 foci, $0.4 \times 10^6$ cells parental and *RAD54L2*-deficient U2OS cells were seeded were seeded onto HCl acid-etched coverslips and cultured in the presence or absence of 4 mM HU (BioBasic) for 24 h before fixation. Cells were washed with ice-cold PBS, and pre-extracted with CSK buffer (300 mM sucrose, 100 mM NaCl, 3 mM MgCl$_2$, 10 mM PIPES pH 7.0, 0.5% (v/v) Triton X-100) for 15 min on ice. Extracted cells were fixed with 4% (w/v) PFA in PBS for 10 min at room temperature, washed three times in PBS, and incubated in blocking buffer (10% goat serum (Thermo Fisher #16210064), 0.5% NP-40, 5% (w/v)

saponin in PBS) 30 min. Blocking buffer containing the appropriate primary antibodies was added to the coverslips for 2 h, the slips were washed three times in PBS, and incubated in blocking buffer containing secondary antibodies and 0.5 µg/mL 4',6-diamidino-2-phenylindole (DAPI; Sigma-Aldrich #D9542) for 1 h, protected from light. After washing with PBS, coverslips were mounted with ProLong Gold Antifade (Thermo Fisher #P36930). Images were acquired on a Zeiss AxioObserver Z1 confocal microscope with a 63x oil-immersion objective lens. Cells that were positive for BLM, MRE11, or RAD51 foci were quantified by manual counting. At least 50 cells per condition were analyzed in each experiment.

For the detection of anaphase bridges, cells were grown on HCl acid-etched coverslips as described above, and mitotic cells were enriched by double thymidine and nocodazole block. Cell extraction and fixation were performed according to previously described protocol for ultrafine bridges (Bizard et al, 2018). Ultrafine bridges were visualized using anti-PICH antibody and Alexa Fluor secondary antibody. Images of anaphase cells were acquired on a Zeiss AxioObserver Z1 confocal microscope.

## Data availability

Mass spectrometry data has been deposited as a complete submission to the MassIVE repository and assigned the accession number MSV000093876 (https://massive.ucsd.edu/ProteoSAFe/dataset.jsp?accession=MSV000093876). Raw and processed source data for all figures is included in the Source Data files.

The source data of this paper are collected in the following database record: biostudies:S-SCDT-10_1038-S44319-025-00374-z.

## Peer review information

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

## Acknowledgements

We thank Dan Durocher for providing cell lines. ACG is a Tier 1 Canada Research Chair in Functional Proteomics. Proteomics work was performed at the Network Biology Collaborative Centre at the Lunenfeld-Tanenbaum Research Institute, a facility supported by the Canada Foundation for Innovation, the Ontarian Government, Genome Canada, and Ontario Genomics (OGI-139). GWB is a Tier I Canada Research Chair in Genome Integrity and was supported by the Canadian Institutes of Health Research (FDN-159913). We are grateful to work on the lands of the Mississaugas of the Credit, the Anishnaabeg, the Haudenosaunee, and the Wendat peoples, land that is now home to many diverse First Nations, Inuit, and Métis peoples.

## Author contributions

**Jung Jennifer Ho**: Formal analysis; Investigation; Methodology; Writing—original draft; Writing—review and editing. **Edith Cheng**: Investigation; Writing—review and editing. **Cassandra J Wong**: Formal analysis; Investigation; Writing—review and editing. **Jonathan R St-Germain**: Formal analysis; Investigation; Writing—review and editing. **Wade H Dunham**: Investigation. **Brian Raught**: Formal analysis; Supervision; Funding acquisition. **Anne-Claude Gingras**: Formal analysis; Supervision; Funding acquisition; Writing—review and editing. **Grant W Brown**: Conceptualization; Formal analysis; Supervision; Funding acquisition; Writing—original draft; Writing—review and editing.

Source data underlying figure panels in this paper may have individual authorship assigned. Where available, figure panel/source data authorship is listed in the following database record: biostudies:S-SCDT-10_1038-S44319-025-00374-z.

## Disclosure and competing interests statement

The authors declare no competing interests.

# Expanded View Figures

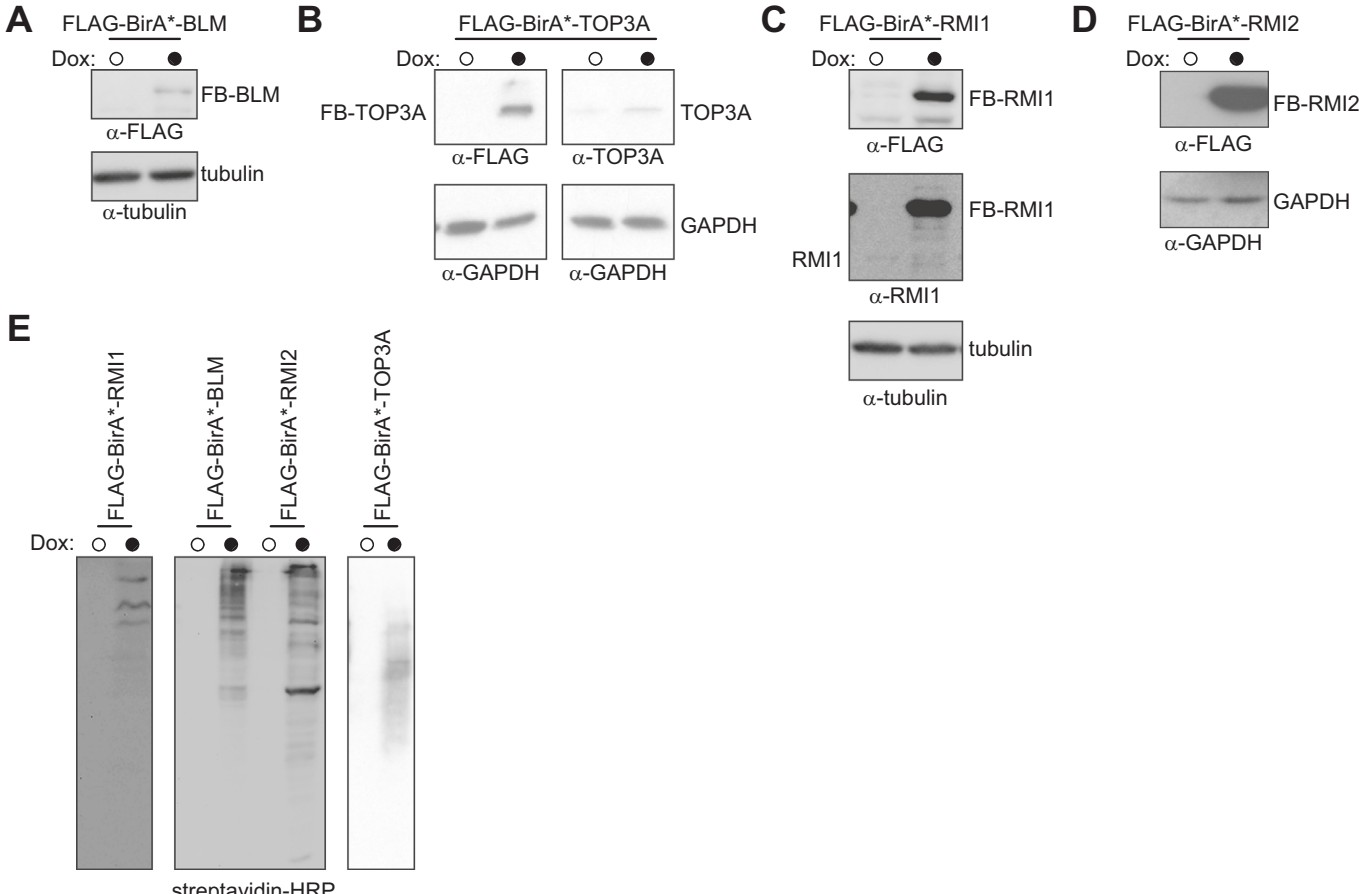

**Figure EV1.  Expression of active BTRR BirA\* fusion proteins.**

(**A**) The stable FLAG-BirA\*-BLM Flp-In T-Rex HEK293 cell line was treated with doxycycline (closed circle) or vehicle (open circle) to induce expression of FLAG-BirA\*-BLM (FB-BLM) and subjected to immunoblot analysis, probing with anti-FLAG or anti-tubulin antibodies, as indicated. (**B**) The stable FLAG-BirA\*-TOP3A Flp-In T-Rex HEK293 cell line was treated with doxycycline (closed circle) or vehicle (open circle) to induce expression of FLAG-BirA\*-TOP3A (FB-TOP3A) and subjected to immunoblot analysis, probing with anti-FLAG, anti-TOP3A, or anti-GAPDH antibodies, as indicated. (**C**) The stable FLAG-BirA\*-RMI1 Flp-In T-Rex HEK293 cell line was treated with doxycycline (closed circle) or vehicle (open circle) to induce expression of FLAG-BirA\*-RMI1 (FB-RMI1) and subjected to immunoblot analysis, probing with anti-FLAG, anti-RMI1, or anti-tubulin antibodies, as indicated. (**D**) The stable FLAG-BirA\*-RMI2 Flp-In T-Rex HEK293 cell line was treated with doxycycline (closed circle) or vehicle (open circle) to induce expression of FLAG-BirA\*-RMI2 (FB-RMI2) and subjected to immunoblot analysis, probing with anti-FLAG, or anti-GAPDH antibodies, as indicated. (**E**) Stable Flp-In T-Rex HEK293 cell lines were treated with doxycycline (closed circle) or vehicle (open circle) to induce expression of the indicated FLAG-BirA\* fusion proteins. Extracts of the cells were fractionated by SDS-PAGE, and biotinylated proteins were detected with streptavidin-HRP. Source data are available online for this figure.

                                                      

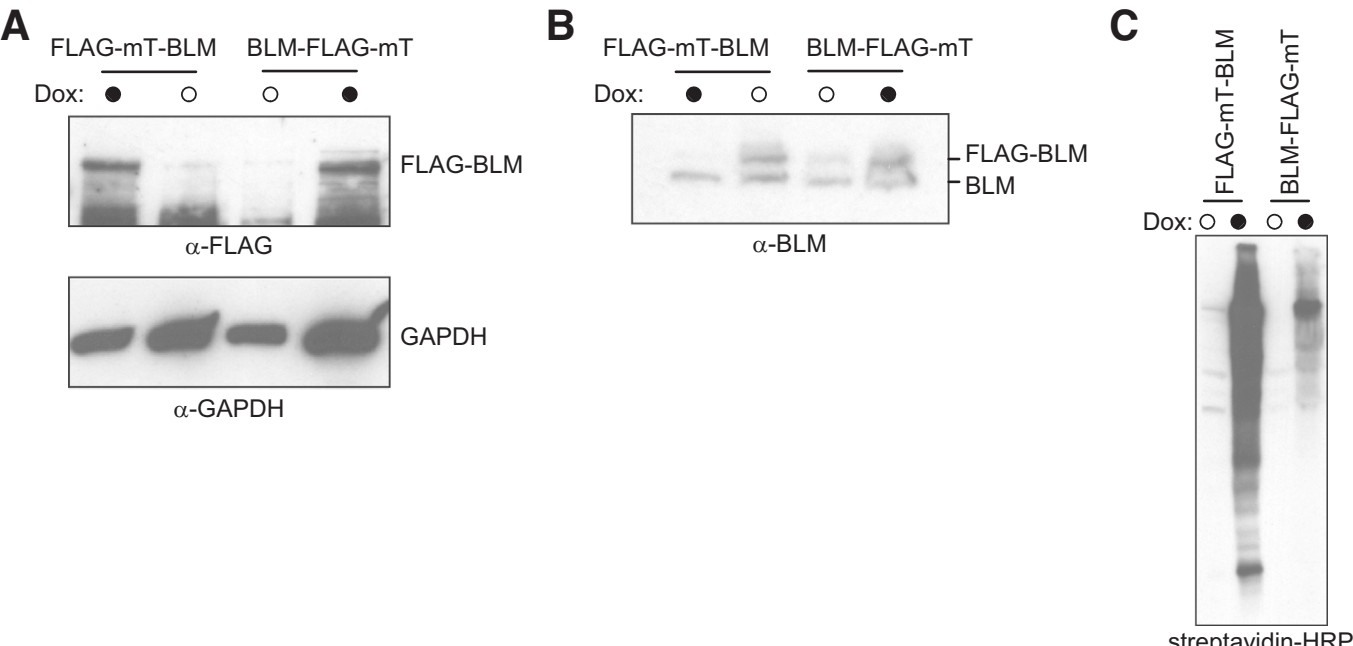

**Figure EV2. Expression of active BLM miniTurbo fusions.**

(A) The stable FLAG-miniTurbo-BLM and BLM-FLAG-miniTurbo cell lines were treated with doxycycline (closed circle) or vehicle (open circle) to induce expression of FLAG-BirA*-BLM (FB-BLM) and subjected to immunoblot analysis, probing with anti-FLAG or anti-GAPDH antibodies, as indicated. (B) The stable FLAG-miniTurbo-BLM and BLM-FLAG-miniTurbo cell lines were treated with doxycycline (closed circle) or vehicle (open circle) to induce expression of FLAG-BirA*-BLM (FB-BLM) and subjected to immunoblot analysis, probing with anti-BLM antibodies. (C) The stable FLAG-miniTurbo-BLM and BLM-FLAG-miniTurbo cell lines were treated with doxycycline (closed circle) or vehicle (open circle) to induce expression of FLAG-BirA*-BLM. Extracts of the cells were fractionated by SDS-PAGE, and biotinylated proteins were detected with streptavidin-HRP. Source data are available online for this figure.

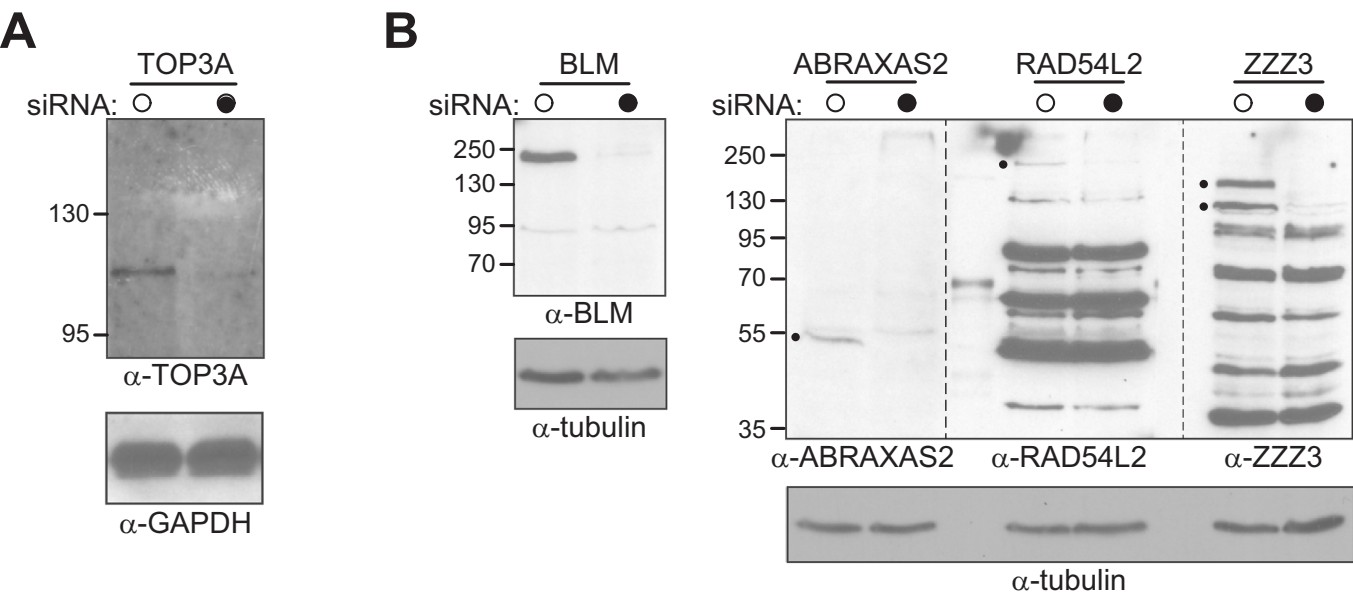

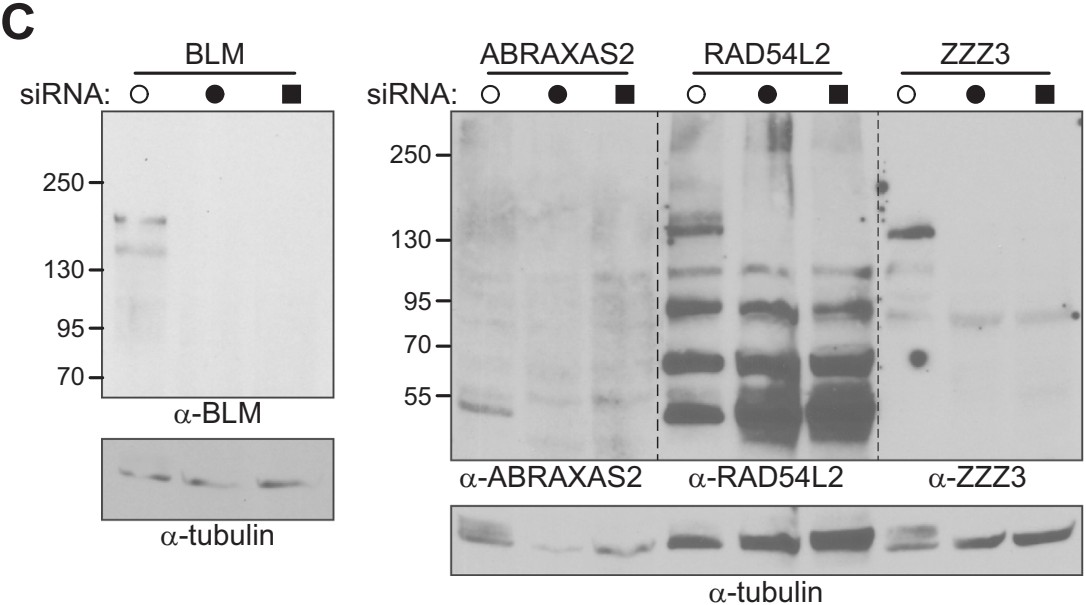

**Figure EV3.  Knockdown of TOP3α, BLM, ABRAXAS2, RAD54L2, and ZZZ3.**

(A) Control (open circles) or *TOP3α* siRNA (closed circles) was transfected into U2OS cells. After 48 h, protein depletion was examined by immunoblot analysis, probing with the antibody against TOP3α or GAPDH (as a loading control). Corresponds to Fig. 3B. (B) Control (open circles) or the indicated siRNAs (closed circles) were transfected into U2OS cells. After 48 h, protein depletion was examined by immunoblot analysis, probing with the indicated antibodies. Anti-tubulin blots are included as loading controls. Small circles mark putative RAD54L2 and ZZZ3 polypeptides. Corresponds to Fig. 3C. (C) Control (open circles), the indicated siRNAs (closed circles), or the indicated siRNAs and the I-SceI expression plasmid (closed squares) were transfected into U2OS DR-GFP cells. After 48 h, protein depletion was examined by immunoblot analysis, probing with the indicated antibodies. Anti-tubulin blots are included as loading controls. Corresponds to Fig. 3D. Source data are available online for this figure.

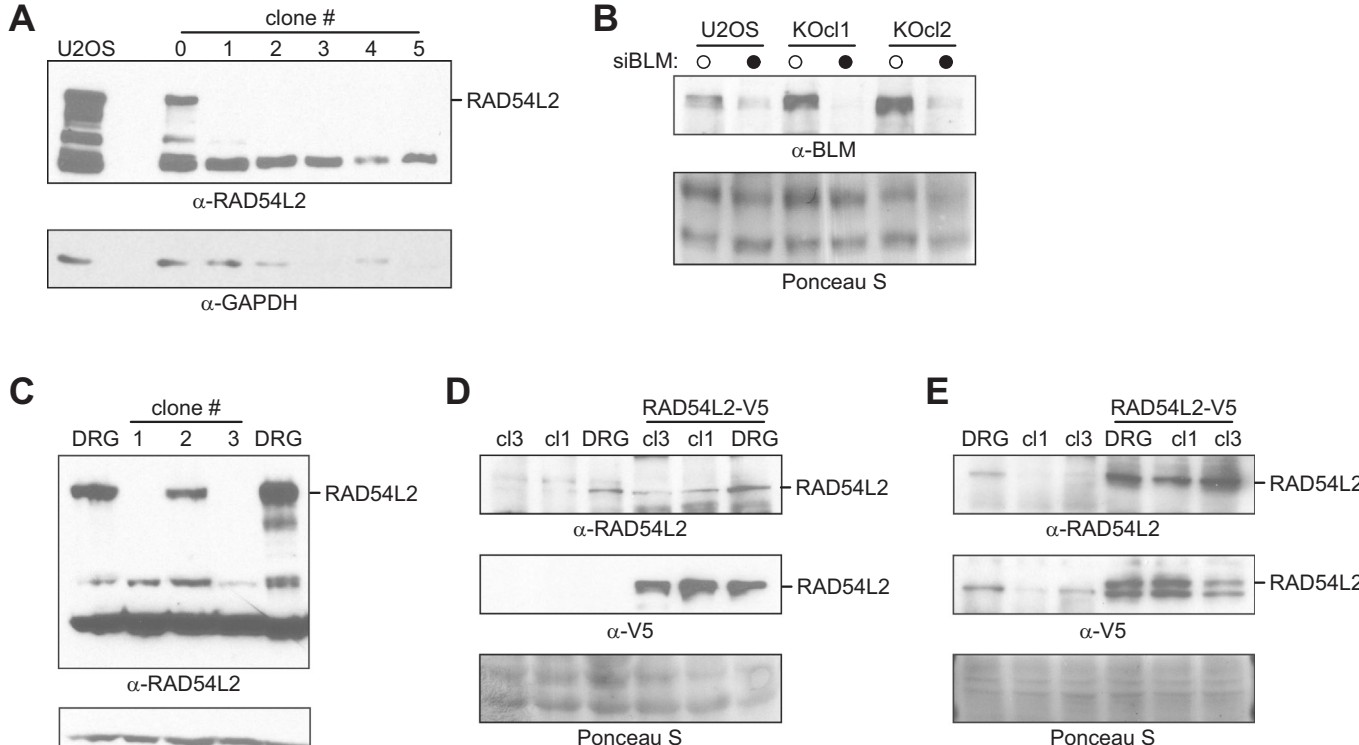

**Figure EV4.  CRISPR/Cas disruption of *RAD54L2*.**

(**A**) U2OS cells were transfected with a plasmid expressing Cas9 and an sgRNA targeting *RAD54L2*, cloned, and examined by immunoblot analysis, probing with antibodies against RAD54L2 or against GAPDH (as a loading control). Parental cells (U2OS) were also examined. The position of RAD54L2 is indicated. Corresponds to Fig. 3E. (**B**). U2OS cells and two *RAD54L2* knockout lines (cl1, cl2) were treated with control (open circles) or *BLM* siRNAs (closed circles), and examined by immunoblot analysis, probing with antibodies against BLM. A section of the membrane stained with Ponceau S is shown as the loading control. Corresponds to Fig. 3E. (**C**) U2OS DR-GFP cells were transfected with a plasmid expressing Cas9 and an sgRNA targeting *RAD54L2*, cloned, and examined by immunoblot analysis, probing with antibodies against RAD54L2 or against tubulin (as a loading control). Parental cells (DRG) were also examined. The position of RAD54L2 is indicated. Corresponds to Fig. 3F. (**D**) U2OS DR-GFP (DRG) and two U2OS DR-GFP *RAD54L2* knockout lines (cl1, cl3) were examined by immunoblotting and probing with the indicated antibodies. A section of the membrane stained with Ponceau S is shown as the loading control. Where indicated, the cells were transfected with a plasmid carrying *RAD54L2* tagged with the V5 epitope (RAD54L2-V5). The position of RAD54L2 is indicated. Corresponds to Fig. 3F. (**E**) As in panel (**D**), for an independent experimental replicate. Corresponds to Fig. 3F. Source data are available online for this figure.

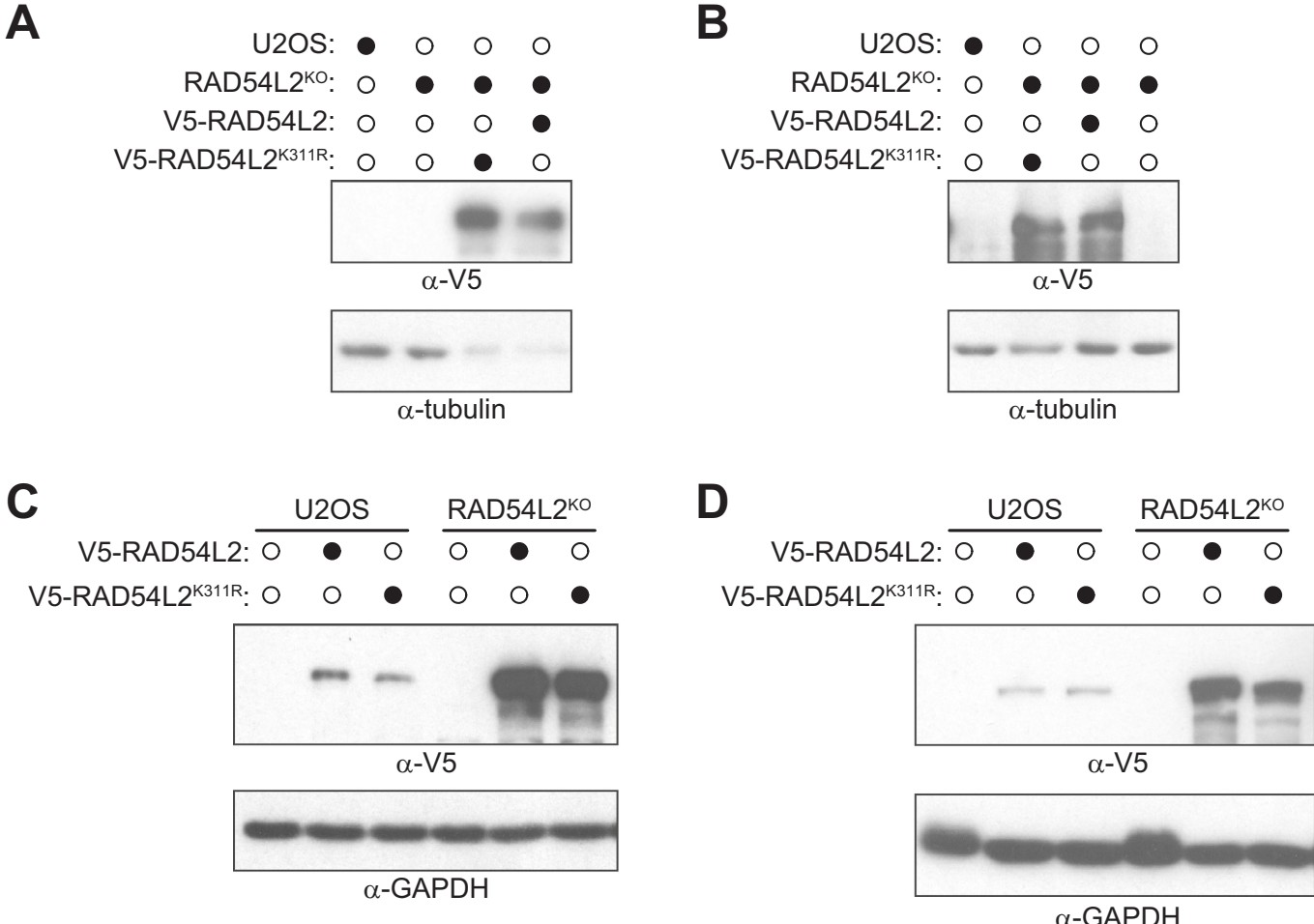

Figure EV5. Rescue of *RAD54L2* knockouts.

(**A**) *RAD54L2*-deficient U2OS cells (clone 1) were mock-transfected (open circles) or transfected with V5-tagged *RAD54L2* or *RAD54L2-K311R* (closed circles). Mock-transfected U2OS cells are also shown. Extracts of the cells were examined by immunoblot analysis, probing with antibodies against V5 or tubulin. Corresponds to Fig. 6A. (**B**) As in panel A, for the independent replicate. Corresponds to Fig. 6A. (**C**) U2OS cells or *RAD54L2*-deficient U2OS cells (clone 1) were mock-transfected (open circles) or transfected with V5-tagged *RAD54L2* or *RAD54L2-K311R* (closed circles). Extracts of the cells were examined by immunoblot analysis, probing with antibodies against V5 or GAPDH. Corresponds to Fig. 6B. (**D**) As in panel (**D**), for the independent replicate. Corresponds to Fig. 6B. Source data are available online for this figure.

