## [Peer Review File · EMBO Reports]

The BLM-TOP3A-RMI1-RMI2 proximity map reveals that RAD54L2 suppresses sister chromatid exchanges

Jung Ho, Edith Cheng, Cassandra Wong, Jonathan St-Germain, Wade Dunham, Brian Raught, Anne-Claude Gingras, and Grant Brown

Corresponding author(s): Grant Brown (grant.brown@utoronto.ca)

Review Timeline:

Submission Date:	7th Apr 24
Editorial Decision:	16th May 24
Revision Received:	21st Nov 24
Editorial Decision:	13th Dec 24
Revision Received:	5th Jan 25
Accepted:	13th Jan 25

Editor: Esther Schnapp

Transaction Report:

Dear Dr. Brown,

Thank you for the submission of your manuscript to EMBO reports. We have now received the full set of referee reports that is pasted below.

As you will see, all referees acknowledge that the findings are interesting. However, they also have several suggestions for how the study could be improved. I think all suggestions are reasonable and should be addressed. Please let me know in case you disagree and we can discuss the exact revision requirements further, also in a video chat, if you like.

I would thus like to invite you to revise your manuscript with the understanding that the referee concerns must be fully addressed and their suggestions taken on board. Please address all referee concerns in a complete point-by-point response. Acceptance of the manuscript will depend on a positive outcome of a second round of review. It is EMBO reports policy to allow a single round of major revision only and acceptance or rejection of the manuscript will therefore depend on the completeness of your responses included in the next, final version of the manuscript.

We realize that it is difficult to revise to a specific deadline. In the interest of protecting the conceptual advance provided by the work, we recommend a revision within 3 months (16th Aug 2024). Please discuss the revision progress ahead of this time with the editor if you require more time to complete the revisions.

- 1) A data availability section providing access to data deposited in public databases is missing. If you have not deposited any data, please add a sentence to the data availability section that explains that.
- 2) Your manuscript contains statistics and error bars based on $n=2$. Please use scatter blots in these cases. No statistics should be calculated if $n=2$.

5) a complete author checklist, which you can download from our author guidelines <<https://www.embopress.org/page/journal/14693178/authorguide>>. Please insert information in the checklist that is also reflected in the manuscript. The completed author checklist will also be part of the RPF.

6) Please note that all corresponding authors are required to supply an ORCID ID for their name upon submission of a revised manuscript (<<https://orcid.org/>>). Please find instructions on how to link your ORCID ID to your account in our manuscript tracking system in our Author guidelines <<https://www.embopress.org/page/journal/14693178/authorguide#authorshipguidelines>>

7) Before submitting your revision, primary datasets produced in this study need to be deposited in an appropriate public database (see <https://www.embopress.org/page/journal/14693178/authorguide#datadeposition>). Please remember to provide a reviewer password if the datasets are not yet public. The accession numbers and database should be listed in a formal "Data Availability" section placed after Materials & Method (see also <https://www.embopress.org/page/journal/14693178/authorguide#datadeposition>). Please note that the Data Availability Section is restricted to new primary data that are part of this study. * Note - All links should resolve to a page where the data can be accessed. *
If your study has not produced novel datasets, please mention this fact in the Data Availability Section.

- the name of the statistical test used to generate error bars and P values,
- the number (n) of independent experiments (please specify technical or biological replicates) underlying each data point,
- the nature of the bars and error bars (s.d., s.e.m.),
- If the data are obtained from $n < 2$, use scatter blots showing the individual data points.

I look forward to seeing a revised form of your manuscript when it is ready.

Referee #1:

In this manuscript, the authors describe the results of extensive proteomics analyses to map the proteins found in proximity to the Bloom syndrome complex (BLM-TOP3A-RMI1/2, or BTR). They identify numerous factors not previously found to interact with BTR components, including RAD54L2. They go on to show that like BLM, RAD54L2 is required to suppress sister chromatid exchanges, and that RAD54L2 plays a role in BLM recruitment to DNA lesions.

This work will be of significant interest to researchers working in the genome stability field. The biotinylation experiments coupled with mass spectrometry to identify proteins in close proximity to BTR are solid, and identify many of the factors previously shown to interact with BLM. The identification of RAD54L2 as a novel player in regulating BTR recruitment is also interesting. Overall, I think this manuscript is a good candidate for publication in EMBO Reports. I have the following suggestions to improve the manuscript prior to publication:

Main points:

1. What is the BLM protein level in RAD54L2 knockout cells? The authors should re-run samples in Fig. EV4B on the same gel. This is important to check in case the apparent recruitment defect is actually due to reduced overall BLM level.
2. Is RAD54L2 present on anaphase bridges with BLM? If so, does it also promote BLM recruitment like it does at DNA damage sites in interphase? If the authors struggle to detect RAD54L2 by immunofluorescence, they could also test this indirectly by quantifying UFBs stained by PICH or RIF1 in RAD54 knockout cells compared to WT and BLM-deficient cells as a control (PICH and RIF1 localise to UFBs independently of BLM).
3. Proximity and co-IP results indicate RAD54L2 may be in the same complex as BLM, but not necessarily through direct protein-protein interactions. Could the authors test using AlphaFold predictions where on BLM, RMI1, RMI2 or TOP3A RAD54L2 may be binding?
4. The model in 5F places RAD54L2 upstream of BTRR before dHJ dissolution, but the authors can't be sure whether it might be acting with BTRR during the dHJ dissolution process. Is the ATPase activity of RAD54L2 required for BLM recruitment in response to HU? If not, then this would more strongly support a model where RAD54L2 and BTR are acting together in a complex at the same step of HR given that the authors have found that the ATPase activity of RAD54L2 suppresses SCEs.

Other points:

1. Original scale bars from microscopy software are still faintly visible in Figure 5 and should be removed.
2. SLX4IP is sometimes referred to as C20orf94, please change all to SLX4IP.
3. Line 54: include Johnson et al., Cancer Res. 2000 who also first showed that BLM binds TOP3A.
4. Line 56: include Xu et al. 2008, who identified RMI2 in back-to-back papers with Singh et al. in G&D.
5. Lines 65-68: include Hodson et al., PNAS 2022 who first showed in vitro that BLM interaction with TOP3A and RMI1 is important for dHJ dissolution.
6. Lines 85-87: This statement is not quite correct. Shorrocks et al. Nat. Comm 2021 showed that mutating the RPA-binding motifs in BLM causes a defect in replication fork restart, but not in other functions of BLM such as SCE suppression.
7. Line 135: Deans and West, Mol. Cell 2009 first identified FANCM as a binding partner for the BTRR complex rather than Lu et al., 2019.
8. Line 136: FAAP24 was identified as a FANCM binding partner by Ciccia et al., Mol. Cell 2007 rather than Coulthard et al., 2013.
9. Line 308: the BLM-MLH1 interaction was also discovered by Langland et al., JBC 2001 at the same time as Pedrazzi et al.
10. Line 315: replace "FA complex" with "FA pathway", given that FANCD2 is the substrate of the FA core complex rather than a stable complex component.

Referee #2:

Ho et al. performed a high-throughput analysis of the BTRR interactome and identified some interesting established and novel interactors of the BLM-TOP3-RMI1-RMI2 complex. They further tested the impact of depleting some of the interactors on sister-chromatid exchange, focusing on ABRAXAS2, ZZZ3, and RAD54L2. Finally, they validated the interaction of RAD54L2 with one of the BTRR components, BLM, and propose that RAD54L2 functions with BLM in promoting non-crossover recombination

events.

Despite the interesting dataset generated, the quality of the follow-up experiments and some of the interpretations drawn from them are questionable. For example, one of the most relevant conclusions from the paper is that RAD54L2 works together with BLM in promoting non-crossover recombination events. However, data in Figure 3E show a further increase in sister-chromatid exchanges when BLM is depleted in RAD54L2 KO cells. Additionally, only the interaction of RAD54L2 with BLM and not with other BTRR components has been validated. Finally, several experiments have been conducted in two replicas only.

In general, the authors could improve the clarity of the manuscript and the figures. Additionally, the following points should be addressed to support the claims made.

Major points:

- Figure 1:
 - The authors validate RAD54L2, which interacts only with BLM. However, other strong candidates interacting with even more than one BTRR complex member, like FAAP24 and ZNF451, should also be validated.
- Figure 3:
 - In general, all the experiments should be conducted in n=3. How many replicas have been conducted for the experiment shown in panel B?
 - This reviewer is surprised to see such a small difference with siTOP3A in panel B. Is the siRNA working? Are the other siRNAs considered not to give a phenotype (WRN, NOC4L, CENPT) working? Otherwise, it would be better not to show the data as they could be misleading as they are now.
 - From data in panel E, the authors conclude that BLM and RAD54L2 are epistatic. However, at least for RAD54L2 KO clone 1, there is a statistically significant increase in sister chromatid exchanges when BLM is depleted. A similar scenario is observed for clone 2, despite not being statistically significant, perhaps because of the number of repeats.
 - The same experimental setup in E should be used to test the non-crossover phenotype.
 - In panel F, what happens when RAD54L2 is overexpressed in BLM depleted cells? Additionally, for this dataset to be reliable, issues regarding the supplementary figure western blot (knock-out validation and overexpression, EV4D/E) should be addressed.
 - The same experimental setup in F should be used to test sister chromatid exchanges.
- Figure 4:
 - Why is RAD54L2 absent from the WCE in panel B? Can the authors show that BLM does not interact with RAD54L2 in the RAD54L2 KO?
 - It would be important to validate RAD54L2 interaction with the other BTRR components as well.
- Figure 5:
 - Considering additional roles of BLM in DNA end resection, RPA foci should also be tested.
- Supplementary Figures
 - o The loading controls are missing in some of the gels (EV4A, EV4B: are the BLM levels reduced in the RAD54L2 KO clone 2 or is it just a matter of loading?)
 - o In EV4D, why is overexpressed RAD54L2 not visible in the WB when blotting with RAD54L2 antibody?
 - o In EV4E, is RAD54L2 KO clone 3 re-expressing RAD54L2? Or is there a mistake in the figure?

Other points:

- Figure 1:
 - This reviewer found it challenging to understand that shared versus unique interactors of each prey protein are plotted in panels A and B, respectively. Can the authors make this clearer in the figure?
 - The dashed lines in panel 3 are supposed to represent the "protein-protein interactions among a subset of the BTRR proximal proteins as annotated in Biogrid". It seems that only interactions involving TP53 are highlighted. It would be useful to show also other interactions between all the proteins in the network.
- Figure 2:
 - In panel B, it would be interesting to compare the turboID results to the BirA results of TOP3A, RMI1, RMI2 from Figure 1. This could also perhaps help understanding the choice of the hits pursued in Figure 3.
- Figure 3:
 - In panel B, the authors select a "subset of proximal proteins from our BTRR BioID screens". However, some of the hits (e.g. ABRAXAS2) are specific to only one of the BTRR components. Others, like ZZZ3, are only interacting with BLM, which has additional functions unrelated to the BTRR complex. The rationale behind the choice of the hits should be made clear and the wording should be changed accordingly.
 - It would be good if the narrative could explain why the authors chose to focus on RAD54L2 and not ABRAXAS2 and/or ZZZ3.
- Figure 4:
 - The data in C should be compared with the interactors of the BTRR complex and not only BLM.
- Figure 5:
 - The authors should show MRE11 foci representative images as well and improve the quality of panel 5A.
 - D'Alessandro et al, 2023 and CRISPR screen data from Olivieri et al, 2020 show that RAD54L2 KO cells are not hypersensitive to hydroxyurea. Olivieri et al, 2020 also show that BLM KO does not hypersensitize cells to HU, perhaps because the crossover pathway can take over. It would be good to read about the authors' perspectives on these issues in the discussion.

Referee #3:

Ho et al performed a proximity biotinylation assay using the BLM-TOP3A-RMI1-RMI2 (BTRR) complex members as baits in order to understand better the mechanism of action of the BTRR complex in preventing crossover formation. Among the different BTRR interacting candidates, they focused on RAD54L2. They show that RAD54L2 co-immunoprecipitate with the BTRR complex and inhibits sister chromatid exchange on its own. Interestingly, the absence of RAD54L2 prevents replication stress-induced BLM foci formation. This supports a model where RDA54L2 promotes BLM recruitment to replication stress-induced recombination intermediate to mature them as noncrossovers.

Overall, the results are of interest for the community and shed light on a novel promising player largely uncharacterized so far.

Major comments

i. By definition, proximity biotinylation identifies proteins that are spatially proximal to the bait, whether they are interacting partners or not. So, this approach identifies the "proximity proteome" but by it does not identify the "proximity interactome". Only part of the proximity proteome may be part of the interactome of a specific bait, and not all the interactome may be identified in the proximity proteome for different reasons. So, the use of "proximity interactome" should be changed all along the manuscript by "proximity proteome". Only when additional experiments are performed can a candidate be considered, or not, as a real interactor. Here, the authors used co-IP to show that RAD54L2 interacts with the BTRR complex. However, the authors should specify that this interaction may not be direct. A simple experiment could be to use DNase or ethidium bromide to test for a potential role of DNA in mediating this interaction.

Related to this comment: line 234-237, the proximity to proteins related to transcription, nucleosome organization etc may simply reflect that DNA damage could occur in the context of transcription but not that RAD54L2 is involved in transcription.

ii. Both in the introduction and in the discussion the authors position BLM either early during the recombination process at the DNA end resection step, or late at the double Holliday junction dissolution step. It is surprising that the authors do not mention the SDSA pathway which is the main homologous recombination pathway in mammalian somatic cells. Before acting on double Holliday junctions, the BTRR likely acts on D-loops to mediate SDSA.

iii. Epistasis between BTRR and RAD54L2 should be better interpreted and presented. It is not possible to say in the same sentence that "BLM and RAD54L2 function in the same genetic pathway... in addition to making independent contributions to SCE suppression." (see lines 297-209). One possibility could be that RAD54L2 facilitates BTRR recruitment to replication accidents, but that some BTRR could be recruited independently of RAD54L2. However, this result is not supported by the complete absence of BLM foci in the absence of RAD54L2. To reconcile all this, one would have to invoke, for instance, that some BLM can be recruited to replication accidents without forming foci.

iv. How come the BTRR complex rely on RAD54L2 to bind to recombination intermediates to process them late while the same BTRR complex is involved in DNA end resection that takes place upstream of the recruitment of RAD51 that appear to be normal in the absence of RAD54L2?

v. This reviewer would have appreciated a series of SCE images as supplementary information to have a sense of the phenotype because the quality of the images presented in Figure 3A is not optimal and the strength of the siBLM SCE phenotype far from the known harlequin phenotype of BLM cell lines. The same applies for the BLM foci which are hardly visible on Figure 5A.

Minor comments

i. Line 64 ... "homology search for a suitable template"

ii. Line 75-76: "random locations" is the opposite of "hotspots". Please refine the description.

iii. May indicate at the end of the introduction that not much is known about RAD54L2, besides 2 recent papers.

Response to Reviewers

We thank the reviewers for providing constructive feedback on our manuscript, as well as their positive remarks and enthusiasm. We have considered each point carefully and have used it to guide our efforts in improving the study in our revised submission. For clarity, we include the comments of each reviewer, followed by our point-by-point responses in BLUE text.

Referee #1:

In this manuscript, the authors describe the results of extensive proteomics analyses to map the proteins found in proximity to the Bloom syndrome complex (BLM-TOP3A-RMI1/2, or BTR). They identify numerous factors not previously found to interact with BTR components, including RAD54L2. They go on to show that like BLM, RAD54L2 is required to suppress sister chromatid exchanges, and that RAD54L2 plays a role in BLM recruitment to DNA lesions.

This work will be of significant interest to researchers working in the genome stability field. The biotinylation experiments coupled with mass spectrometry to identify proteins in close proximity to BTR are solid, and identify many of the factors previously shown to interact with BLM. The identification of RAD54L2 as a novel player in regulating BTR recruitment is also interesting. Overall, I think this manuscript is a good candidate for publication in EMBO Reports. I have the following suggestions to improve the manuscript prior to publication:

Main points:

1. What is the BLM protein level in RAD54L2 knockout cells? The authors should re-run samples in Fig. EV4B on the same gel. This is important to check in case the apparent recruitment defect is actually due to reduced overall BLM level.

We have re-run the samples on the same gel and revised Fig. EV4B accordingly. The level of BLM is not decreased in the RAD54L2 ko cells.

2. Is RAD54L2 present on anaphase bridges with BLM? If so, does it also promote BLM recruitment like it does at DNA damage sites in interphase? If the authors struggle to detect RAD54L2 by immunofluorescence, they could also test this indirectly by quantifying UFBs stained by PICH or RIF1 in RAD54 knockout cells compared to WT and BLM-deficient cells as a control (PICH and RIF1 localise to UFBs independently of BLM).

We were unable to detect RAD54L2 on anaphase bridges (example image shown below). However, we do not know if the anti-RAD54L2 antibody performs well in this application.

As suggested, we stained mitoses for PICH and found that the RAD54L2 knockout accumulates PICH-positive ultra-fine bridges. The data are now included in Figure 6.

3. Proximity and co-IP results indicate RAD54L2 may be in the same complex as BLM, but not necessarily though direct protein-protein interactions. Could the authors test using AlphaFold predictions where on BLM, RMI1, RMI2 or TOP3A RAD54L2 may be binding?

No interface was predicted with AlphaFold using the following pairwise combinations:

Pair	interface	avg_models	Residue contacts
BLM / RAD54L2	A:B	0	0
TOP3A/ RAD54L2	A:B	0	0
RMI1 / RAD54L2	A:B	0	0
RMI2/ RAD54L2	A:B	0	0

4. The model in 5F places RAD54L2 upstream of BTRR before dHJ dissolution, but the authors can't be sure whether it might be acting with BTRR during the dHJ dissolution process. Is the ATPase activity of RAD54L2 required for BLM recruitment in response to

HU? If not, then this would more strongly support a model where RAD54L2 and BTR are acting together in a complex at the same step of HR given that the authors have found that the ATPase activity of RAD54L2 suppresses SCEs.

We agree with the reviewer that BLM recruitment and dissolution could have different requirements for RAD54L2. As suggested, we tested if the ATPase activity of RAD54L2 is required for BLM recruitment during HU treatment. We found that ATPase-dead RAD54L2 did not rescue BLM focus formation in HU, suggesting that RAD54L2 functions upstream of BTRR. The new data are included in Figure 6.

Other points:

1. Original scale bars from microscopy software are still faintly visible in Figure 5 and should be removed.

Original scale bars have been removed from Fig 5.

2. SLX4IP is sometimes referred to as C20orf94, please change all to SLX4IP.

C20orf94 has been changed to SLX4IP in Figure 2.

3. Line 54: include Johnson et al., Cancer Res. 2000 who also first showed that BLM binds TOP3A.

Johnson et al. added to line 54.

4. Line 56: include Xu et al. 2008, who identified RMI2 in back-to-back papers with Singh et al. in G&D.

Xu et al. added to line 56.

5. Lines 65-68: include Hodson et al., PNAS 2022 who first showed in vitro that BLM interaction with TOP3A and RMI1 is important for dHJ dissolution.

Hodson et al. added to line 66.

6. Lines 85-87: This statement is not quite correct. Shorrocks et al. Nat. Comm 2021 showed that mutating the RPA-binding motifs in BLM causes a defect in replication fork restart, but not in other functions of BLM such as SCE suppression.

Statement edited and reference inserted. See line 88.

7. Line 135: Deans and West, Mol. Cell 2009 first identified FANCM as a binding partner for the BTRR complex rather than Lu et al., 2019.

Deans & West added to line 135.

8. Line 136: FAAP24 was identified as a FANCM binding partner by Ciccina et al., Mol. Cell 2007 rather than Coulthard et al., 2013.

Ciccina et al. added to line 137.

9. Line 308: the BLM-MLH1 interaction was also discovered by Langland et al., JBC 2001 at the same time as Pedrazzi et al.

Langland et al. added to line 309.

10. Line 315: replace "FA complex" with "FA pathway", given that FANCD2 is the substrate of the FA core complex rather than a stable complex component.

Corrected, line 321.

Referee #2:

Ho et al. performed a high-throughput analysis of the BTRR interactome and identified some interesting established and novel interactors of the BLM-TOP3-RMI1-RMI2 complex. They further tested the impact of depleting some of the interactors on sister-chromatid exchange, focusing on ABRAXAS2, ZZZ3, and RAD54L2. Finally, they validated the interaction of RAD54L2 with one of the BTRR components, BLM, and propose that RAD54L2 functions with BLM in promoting non-crossover recombination events.

Despite the interesting dataset generated, the quality of the follow-up experiments and some of the interpretations drawn from them are questionable. For example, one of the most relevant conclusions from the paper is that RAD54L2 works together with BLM in promoting non-crossover recombination events. However, data in Figure 3E show a further increase in sister-chromatid exchanges when BLM is depleted in RAD54L2 KO cells. Additionally, only the interaction of RAD54L2 with BLM and not with other BTRR components has been validated. Finally, several experiments have been conducted in two replicas only.

In general, the authors could improve the clarity of the manuscript and the figures. Additionally, the following points should be addressed to support the claims made.

Major points:

Figure 1 The authors validate RAD54L2, which interacts only with BLM. However, other strong candidates interacting with even more than one BTRR complex member, like FAAP24 and ZNF451, should also be validated.

Our downstream analysis focussed on RAD54L2. FAAP24 is a member of the FA complex (PMID: 17289582) whose interaction with BTRR is well-established (PMID: 12724401, PMID: 20064461, PMID: 15257300). ZNF451 is certainly of interest, given the functional relationship with RAD54L2 that has been recently described. However, we feel that SCE assays of additional BTRR proximal proteins is outside of the scope of the current analysis.

Figure 3 In general, all the experiments should be conducted in n=3. How many replicas have been conducted for the experiment shown in panel B?

Figure 3B is a single replicate, as it represents a low-throughput secondary screen for proximal proteins that function in the resolution of sister chromatids. RAD54L2, ABRAXAS2, and ZZZ3 were assayed for SCEs two more times (Figure 3C) and twice for non-crossover recombination (Figure 3D). The data clearly support our inference that depletion of RAD54L2, ABRAXAS2, and ZZZ3 results in increased SCEs.

This reviewer is surprised to see such a small difference with siTOP3A in panel B. Is the siRNA working? Are the other siRNAs considered not to give a phenotype (WRN, NOC4L, CENPT) working? Otherwise, it would be better not to show the data as they could be misleading as they are now.

Depletion of TOP3A was validated by immunoblot, now included in Figure EV3. Perhaps the depletion of TOP3A was not sufficient to yield a greater increase in SCEs. We now note that the effect of siTOP3A is modest, on line 188.

Depletion of each protein in the secondary screen was not assessed, and so WRN, NOC4L, and CENPT could be false negatives in the SCE assay. We have noted this possibility on line 192.

From data in panel E, the authors conclude that BLM and RAD54L2 are epistatic. However, at least for RAD54L2 KO clone 1, there is a statistically significant increase in sister chromatid exchanges when BLM is depleted. A similar scenario is observed for clone 2, despite not being statistically significant, perhaps because of the number of repeats.

Our reasoning is that if BLM and RAD54L2 display negative epistasis or synergistic epistasis then the expectation is that the combination of siBLM and RAD54L2 knockout should be additive or multiplicative when compared to the single depletions. The effect that we observed was less than additive for both RAD54L2 knockout clones and so we infer that BLM and RAD54L2 are not entirely independent for the SCE phenotype. As the reviewer notes, there is some increase in SCEs, suggesting that BLM and RAD54L2 have some independence. We have modified the wording on lines 210 – 212 to indicate that the data are consistent with BLM and RAD54L2 functioning at least partially in the same pathway to suppress SCEs.

The same experimental setup in E should be used to test the non-crossover phenotype.

The experiment has been repeated, as suggested, and Figure 3F has been updated accordingly. We now include both RAD54L2 knockout lines with the DR-GFP reporter and rescue the NCO defect by expressing RAD54L2. We also find that the combination of siBLM and RAD54L2 knockout has %NCO (1.33%) very similar to siBLM alone (1.12%), suggesting that BLM and RAD54L2 are in the same pathway to promote NCO recombination. The text on lines 216 – 222 has been amended to reflect the new data.

In panel F, what happens when RAD54L2 is overexpressed in BLM depleted cells?

We were not able to recover enough cells for analysis of RAD54L2 overexpression in BLM knockdown cells because of poor cell growth following the triple transfection (siBLM, I-SceI plasmid, and RAD54L2 plasmid).

Additionally, for this dataset to be reliable, issues regarding the supplementary figure western blot (knock-out validation and overexpression, EV4D/E) should be addressed.

We have repeated the immunoblots for two of the replicates in panel F, and have amended Figure EV4 D and E, as suggested.

The same experimental setup in F should be used to test sister chromatid exchanges.

We showed that expression of RAD54L2 rescues the increased SCE phenotype of the RAD54L2 knockout Figure 5E. That data now appears in Figure 6.

Figure 4

Why is RAD54L2 absent from the WCE in panel B? Can the authors show that BLM does not interact with RAD54L2 in the RAD54L2 KO?

The same amount of protein is loaded for WCE and nuclear extract, resulting in an enrichment of nuclear proteins. RAD54L2 is not detected in the given micrograms of WCE but is detected in the same number of micrograms of nuclear extract. To avoid confusion we have removed the WCE lanes from the panel.

The co-IP in Figure 4B was performed with the BirA*-FLAG-BLM HEK293T cell line used for BioID. We have not knocked out RAD54L2 in this line. The specificity control for the co-IP is in the right-most lane, where if we do not induce expression of BirA*-FLAG-BLM we do not detect RAD54L2 in the immunoprecipitate. We have revised the figure legend and labels to clarify. We also performed a similar co-IP, purifying miniTurbo-BLM by virtue of its FLAG epitope tag, and again found that RAD54L2 was present in the immunoprecipitate only when expression of miniTurbo-BLM was induced. These data are now in Figure 4C, and we have revised the text on lines 231 – 238.

It would be important to validate RAD54L2 interaction with the other BTRR components as well.

We don't feel that testing TOP3 α , RMI1, and RMI2 interactions with RAD54L2 is important, given that none of these proteins were prominent in the RAD54L2 BioID, and that RAD54L2 was not identified in the TRR BioID's.

Figure 5

Considering additional roles of BLM in DNA end resection, RPA foci should also be tested.

MRN was chosen as a proxy for initial resection. RPA foci could indicate ssDNA at sites other than DSBs.

Supplementary Figures

The loading controls are missing in some of the gels (EV4A, EV4B: are the BLM levels reduced in the RAD54L2 KO clone 2 or is it just a matter of loading?)

Loading controls have been added to EV4A and for EV4B we have re-run the samples on the same gel. The level of BLM is not decreased in the RAD54L2 ko lines.

In EV4D, why is overexpressed RAD54L2 not visible in the WB when blotting with RAD54L2 antibody?

We have repeated the westerns in EV4D and EV4E, running more lysate to improve the detection of RAD54L2. We now are able to detect the overexpressed RAD54L2.

In EV4E, is RAD54L2 KO clone 3 re-expressing RAD54L2? Or is there a mistake in the figure?

There might have been spill-over from the adjacent lane. The blot has been repeated, and we do not see evidence of RAD54L2 expression.

Other points:

Figure 1

This reviewer found it challenging to understand that shared versus unique interactors of each prey protein are plotted in panels A and B, respectively. Can the authors make this clearer in the figure?

We have added labels to Figure 1 to clarify.

The dashed lines in panel 3 are supposed to represent the "protein-protein interactions among a subset of the BTRR proximal proteins as annotated in Biogrid". It seems that only interactions involving TP53 are highlighted. It would be useful to show also other interactions between all the proteins in the network.

We chose to highlight only the TP53 network to prevent the figure from becoming too busy.

Figure 2:

In panel B, it would be interesting to compare the turboID results to the BirA results of TOP3A, RMI1, RMI2 from Figure 1. This could also perhaps help understanding the choice of the hits pursued in Figure 3.

We have of course made this comparison when exploring the data, but we didn't find a compelling reason to include it.

Figure 3:

In panel B, the authors select a "subset of proximal proteins from our BTRR BioID screens". However, some of the hits (e.g.ABRAXAS2) are specific to only one of the BTRR components. Others, like ZZZ3, are only interacting with BLM, which has additional functions unrelated to the BTRR complex. The rationale behind the choice of the hits should be made clear and the wording should be changed accordingly.

It would be good if the narrative could explain why the authors chose to focus on RAD54L2 and not ABRAXAS2 and/or ZZZ3.

We didn't have a particularly strong rationale for the candidates we chose. We tried to pick mostly unknowns with interesting functions. We have inserted a sentence (line 201) describing our focus on RAD54L2, which was based on it being a SNF2 ATPase with a high degree of similarity to RADX, a known recombination modulator.

Figure 4:

The data in C should be compared with the interactors of the BTRR complex and not only BLM.

We chose to make the comparison to BLM, since RAD54L2 was not identified in BioID with the other BTRR subunits. All of the BioID data is provided should anyone want to make additional comparisons.

Figure 5:

The authors should show MRE11 foci representative images as well and improve the quality of panel 5A.

We have increased the contrast in 5A to better show the BLM foci, and have added insets to the merge images. Representative images for MRE11 foci and RAD51 foci are now included in Figure 5.

D'Alessandro et al, 2023 and CRISPR screen data from Olivieri et al, 2020 show that RAD54L2 KO cells are not hypersensitive to hydroxyurea. Olivieri et al, 2020 also show that BLM KO does not hypersensitize cells to HU, perhaps because the crossover pathway can take over. It would be good to read about the authors' perspectives on these issues in the discussion.

We agree that differences between overt sensitivity and repair deficits are very interesting, but we don't think we are in a position to comment on the screen results of Olivieri et al, since we have not directly assessed HU sensitivity.

Referee #3:

Ho et al performed a proximity biotinylation assay using the BLM-TOP3A-RMI1-RMI2 (BTRR) complex members as baits in order to understand better the mechanism of action of the BTRR complex in preventing crossover formation. Among the different BTRR interacting candidates, they focused on RAD54L2. They show that RAD54L2 co-immunoprecipitate with the BTRR complex and inhibits sister chromatid exchange on its own. Interestingly, the absence of RAD54L2 prevents replication stress-induced BLM foci formation. This supports a model where RDA54L2 promotes BLM recruitment to replication stress-induced recombination intermediate to mature them as noncrossovers.

Overall, the results are of interest for the community and shed light on a novel promising player largely uncharacterized so far.

Major comments

1. By definition, proximity biotinylation identifies proteins that are spatially proximal to the bait, whether they are interacting partners or not. So, this approach identifies the "proximity proteome" but it does not identify the "proximity interactome". Only part of the proximity proteome may be part of the interactome of a specific bait, and not all the interactome may be identified in the proximity proteome for different reasons. So, the use of "proximity interactome" should be changed all along the manuscript by "proximity proteome". Only when additional experiments are performed can a candidate be considered, or not, as a real interactor. Here, the authors used co-IP to show that RAD54L2 interacts with the BTRR complex. However, the authors should specify that this interaction may not be direct. A simple experiment could be to use DNase or ethidium bromide to test for a potential role of DNA in mediating this interaction.

We agree with the reviewer and have changed the terminology to proximal proteome throughout. We also indicate that the co-IP of RAD54L2 with BLM could be indirect (line 240), for clarity.

Related to this comment: line 234-237, the proximity to proteins related to transcription, nucleosome organization etc may simply reflect that DNA damage could occur in the context of transcription but not that RAD54L2 is involved in transcription.

Roles of RAD54L2 in transcriptional regulation have been described. A clarifying comment is added on line 251.

Both in the introduction and in the discussion the authors position BLM either early during the recombination process at the DNA end resection step, or late at the double Holliday junction dissolution step. It is surprising that the authors do not mention the SDSA pathway which is the main homologous recombination pathway in mammalian somatic cells. Before acting on double Holliday junctions, the BTRR likely acts on D-loops to mediate SDSA.

We agree that it is possible that RAD54L2 plays a role in SDSA along with BLM to affect SCE frequency. However, we have not assayed RAD54L2 knockouts with an SDSA reporter, and the RAD54L2 proximal proteome did not include SDSA-promoting proteins such as FANCM, RTEL1, or RECQ5. As such, we elected to not discuss SDSA.

Epistasis between BTRR and RAD54L2 should be better interpreted and presented. It is not possible to say in the same sentence that "BLM and RAD54L2 function in the same genetic pathway... in addition to making independent contributions to SCE suppression." (see lines 297-209). One possibility could be that RAD54L2 facilitates BTRR recruitment to replication accidents, but that some BTRR could be recruited independently of RAD54L2. However, this result is not supported by the complete absence of BLM foci in the absence of RAD54L2. To reconcile all this, one would have to invoke, for instance, that some BLM can be recruited to replication accidents without forming foci.

We have clarified the presentation of the genetic interaction, as indicated in our response to Reviewer 1:

Our reasoning is that if BLM and RAD54L2 display negative epistasis or synergistic epistasis then the expectation is that the combination of siBLM and RAD54L2 knockout

should be additive or multiplicative when compared to the single depletions. The effect that we observed was less than additive for both RAD54L2 knockout clones and so we infer that BLM and RAD54L2 are not entirely independent for the SCE phenotype. As the reviewer notes, there is some increase in SCEs, suggesting that BLM and RAD54L2 have some independence. We have modified the wording on lines 210 – 212 to indicate that the data are consistent with BLM and RAD54L2 functioning at least partially in the same pathway to suppress SCEs.

How come the BTRR complex rely on RAD54L2 to bind to recombination intermediates to process them late while the same BTRR complex is involved in DNA end resection that takes place upstream of the recruitment of RAD51 that appear to be normal in the absence of RAD54L2?

TOP3A, RMI1, and RMI2 do not appear to be required for the resection carried out by BLM and DNA2. The early and late functions of BLM are likely distinct.

This reviewer would have appreciated a series of SCE images as supplementary information to have a sense of the phenotype because the quality of the images presented in Figure 3A is not optimal and the strength of the siBLM SCE phenotype far from the known harlequin phenotype of BLM cell lines. The same applies for the BLM foci which are hardly visible on Figure 5A.

BLM patient lines are known to have more SCEs than BLM knockdown, with patient lines showing ~10-fold increase and knockdowns showing less. For example, SCEs quantified by strand-seq are lower for BLM knockout (7.76 per cell) and WT (3.32 per cell) [PMID: 35483548; KBM-7 cells] than they are for BLM patient lines (39 per cell) [PMID: 29348659]. Conventional detection on mitotic spreads yielded a ~6x increase in BLM Δ HCT116 cells [PMID: 38266639] and a ~5x increase in siBLM HeLa cells [PMID: 35119917]. The increases in SCEs that we see for siBLM in U2OS are somewhat less (3-4x in Figure 3B, C, E), possibly due to the cell line or the siRNA used.

We have inverted the chromosome spread images and adjusted the contrast to make the sister chromatids, and the exchanges, more visible. We include in the source data for Figure 5A the raw inverted images and the adjusted images, and the phenotypes are accurately quantified in panels B, C, and E.

As noted in our response to Reviewer 2, we have increased the contrast in Fig 5A to better reveal the BLM foci.

Minor comments:

- Line 64 ... "homology search for a suitable template"
Changed to 'homology search'
- Line 75-76: "random locations" is the opposite of "hotspots". Please refine the description.
Changed to indicate that SCE's occur genome-wide, and that there are hotspots.
- May indicate at the end of the introduction that not much is known about RAD54L2, besides 2 recent papers.
Left as is.

Dear Dr. Brown,

Thank you for the submission of your revised manuscript. We have now received the enclosed reports from the referees that were asked to assess it, and I am happy to say that all referees support its publication now. Both referees 1 and 2 still have a few more suggestions that I would like you to incorporate before we can proceed with the official acceptance of your manuscript. Please co-submit a point-by-point response to all final comments.

I agree with referee 2 that ideally all experiments should be based on at least 3 independent replicates.

A few editorial requests will also need to be addressed :

- Please add up to 5 keywords to the ms file.
- Please correct the conflict of interest subheading to "Disclosure Statement and Competing Interests"
- The author credits need to be removed from the ms file. All credits need to be entered during online ms submission.
- The EV figures are not part of the main ms file and only need to be uploaded as individual figures.
- A callout for Fig 4E is missing, please add.
- For the Reagents & Tools TABLE, please use our latest Word template that can be downloaded from our author guidelines (under Structured Methods):
<https://www.embopress.org/page/journal/14693178/authorguide#manuscriptpreparation>
- Source Data (SD) for the main figures need to be uploaded as 1 figure per folder and inside each folder, the files should be organized in subfolders/files, one subfolder/file for each panel. SD for all EV figures can be grouped into one zipped folder.
- Materials and Methods should be Methods
- Please add the specific URL(s) for the deposited dataset(s) provided in the data availability section.
- Please indicate the statistical test used for data analysis in the legends of figures 1d; 2c; 4e.
- Please note that scale bar and its definition are missing for figure 6c.

EMBO press papers are accompanied online by A) a short (1-2 sentences) summary of the findings and their significance, B) 2-3 bullet points highlighting key results and C) a synopsis image that is exactly 550 pixels wide and 200-600 pixels high (the height is variable). The synopsis image should provide a sketch of the major findings, like a graphical abstract. Please note that text needs to be readable at the final size. Please send us this information along with the final manuscript.

Referee #1:

The authors have addressed most of my points, except that they did not address whether BLM recruitment to UFBs requires RAD54L2 (main point 2 in my original review). However, one could argue that could be addressed in future studies, and their manuscript is otherwise much improved. So I think it is now almost ready for publication, if they could address the two minor points below:

1. The inclusion of immunoblots to assess the level of BLM in RAD54L2 knockout cells is welcome (main point 1 in my original

review). However, it appears that BLM is significantly upregulated in RAD54L2 knockout cells. It is impossible to be sure of this unfortunately, because the tubulin loading control is so over-exposed as to be experimentally meaningless. Could the authors repeat this, and if it still appears that BLM is upregulated, comment/speculate on why this might be in the manuscript in an appropriate place.

2. Line 76: the Hamadeh et al. reference is not in the reference list.

Referee #2:

Ho et al. have improved their manuscript by demonstrating the accumulation of PICH-positive ultra-fine anaphase bridges in RAD54L2-deficient cells, and by incorporating additional controls to enhance the reliability of their figures. However, several points raised in the initial review remain unaddressed. Below, I outline specific concerns and recommendations:

Major Points:

- Figure 5: RPA foci as markers of long resection. RPA foci were suggested as an indication of long resection (which is the step where BLM is involved), MRN loading does not necessarily correlate with long resection, so the fact that MRN binding is not affected does not imply that the proper resection is not affected neither. Therefore, the authors cannot conclude from this experiment that RAD54L2 is not required for BLM's role in resection.
- The discussion would benefit from addressing the observed differences between the lack of hypersensitivity in RAD54L2 KO cells upon HU treatment observed in other studies and the repair phenotypes described. Even without performing sensitivity assays, speculating on the potential implications of these observations would enrich the narrative.
- A critical requirement for any scientific publication is the inclusion of at least n=3 biological replicates for all experiments, regardless of how strong the observed phenotype may appear. Exceptions can be made for screening experiments, provided they are followed by subsequent validation experiments, which should also meet the n=3 criteria. In this reviewer's opinion, meeting this condition is essential for the manuscript to be considered for publication in EMBO Reports, but it may be the journal/editor's discretion whether n =2 experiments are sufficient. This recommendation applies to Figures 3C, 3D, 3E, and 6B. Additionally, proper statistical analyses are required to support the conclusions drawn.

Minor Points and Corrections:

- Line 134: Figure 1A should also be referred to.
- Figures 1A and 1C: These panels present overlapping information and should be consolidated into a single panel for clarity.
- Figure 1F: Overexpression of RAD54L2 in siBLM-deficient cells was suggested by this reviewer. The authors faced challenges in recovering cells after transfection with siBLM, RAD54L2 plasmid and I-SceI plasmid simultaneously. A more practical approach would have been to first generate a cell line stably expressing RAD54L2 (e.g., via a lentiviral vector) before introducing additional experimental conditions. This strategy would have simplified the system.
- Lines 162-164: as far as I understand, in Fig1A/2B, PML is shown to be detected in BirA-BLM (N-terminus tagged), therefore I do not think that they authors can claim "One exception was PML, which was detected with BirA*-TOP3A, BirA*-RMI1, and BLM-mT, suggesting that PML is not in close proximity to the BLM amino terminus".
- Figures 2A, 2B, and 4D: "proteins identified with {greater than or equal to}2 baits are shown" should be specified, as stated in Figure 1A, to maintain the clarity.
- Line 191: Statistical significance cannot be claimed with n=1; cannot be stated as 'resulted in a statistically supported increase'.
- Figure EV4A: The loading control is not optimal. Since the authors are also showing the TIDE analysis for these clones, I find this a minor point; but from this blot alone one cannot conclude if the KOs are reliable.
- Figures 4B and 4C: Ideally, an empty mt-FLAG vector control should be included to confirm interaction specificity. However, given the corroborating data between experiments, this is not critical.
- Line 242: Refer to Figure 4D, not Figure 4C.
- Line 245: Refer to Figure 4D, not Figure 4C.
- Line 248: Refer to Figure 4E, not Figure 4D.
- Line 251: Refer to Figure 4E, not Figure 4D.
- HU treatment details: Time and concentration for HU treatment should be included in the Results section of the manuscript or in the figure legend.
- Figure EV5D legend: The legend should state "As in panel C" instead of "As in panel D."
- Figures 6C and 6D: Since these panels derive from the same experiment, they should be presented together in the same panel.
- Line 303: The data do not conclusively determine whether RAD54L2 is required for processing ultra-fine bridges or preventing their formation. The possibility of defects in decatenation by TOP2 cannot be excluded, particularly since no HU treatment was applied in this experiment. Including data on the combined effect of siBLM and RAD54L2 KO would clarify this point. But at least "processing" should be changed to "avoiding accumulation".

Referee #3:

The authors addressed properly all my concerns

We appreciate the additional comments provided by the editor and reviewers, and their efforts to improve our manuscript. Our point-by-point responses to the editorial requests and reviewers' comments follow:

Editorial Requests:

- Please add up to 5 keywords to the ms file.

Added

- Please correct the conflict of interest subheading to "Disclosure Statement and Competing Interests"

Corrected

- The author credits need to be removed from the ms file. All credits need to be entered during online ms submission.

Removed

- The EV figures are not part of the main ms file and only need to be uploaded as individual figures.

Noted

- A callout for Fig 4E is missing, please add.

Added, and callouts for other Fig 4 panels corrected

- For the Reagents & Tools TABLE, please use our latest Word template that can be downloaded from our author guidelines (under Structured Methods):

New template used

- Source Data (SD) for the main figures need to be uploaded as 1 figure per folder and inside each folder, the files should be organized in subfolders/files, one subfolder/file for each panel. SD for all EV figures can be grouped into one zipped folder.

SD has been reorganized accordingly

- Materials and Methods should be Methods

Corrected

- Please add the specific URL(s) for the deposited dataset(s) provided in the data availability section.

URL has been added, and datasets are now public

- Please indicate the statistical test used for data analysis in the legends of figures 1d; 2c; 4e.

Indicated in the Figure legends

- Please note that scale bar and its definition are missing for figure 6c.

Scale bar added and defined

Referee #1:

The authors have addressed most of my points, except that they did not address whether BLM recruitment to UFBs requires RAD54L2 (main point 2 in my original review). However, one could argue that could be addressed in future studies, and their manuscript is otherwise much improved. So I think it is now almost ready for publication, if they could address the two minor points below:

1. The inclusion of immunoblots to assess the level of BLM in RAD54L2 knockout cells is welcome (main point 1 in my original review). However, it appears that BLM is significantly upregulated in RAD54L2 knockout cells. It is impossible to be sure of this unfortunately, because the tubulin loading control is so over-exposed as to be experimentally meaningless. Could the authors repeat this, and if it still appears that BLM is upregulated, comment/speculate on why this might be in the manuscript in an appropriate place.

We have replaced the tubulin blot with an image of the Ponceau S stained membrane. It was not possible to dial down the tubulin signal given that we had to load enough protein to detect BLM, which is present in low abundance. We added a comment that BLM abundance increases (lines 213-215), but prefer not to speculate on why this might be.

2. Line 76: the Hamadeh et al. reference is not in the reference list.

Reference added.

Referee #2:

Ho et al. have improved their manuscript by demonstrating the accumulation of PICH-positive ultra-fine anaphase bridges in RAD54L2-deficient cells, and by incorporating additional controls to enhance the reliability of their figures. However, several points raised in the initial review remain unaddressed. Below, I outline specific concerns and recommendations:

Major Points:

- Figure 5: RPA foci as markers of long resection. RPA foci were suggested as an indication of long resection (which is the step where BLM is involved), MRN loading does not necessarily correlate with long resection, so the fact that MRN binding is not affected does not imply that the proper resection is not affected neither. Therefore, the authors cannot conclude from this experiment that RAD54L2 is not required for BLM's role in resection.

Since RPA foci can also indicate ssDNA from sources distinct from double strand break resection, we chose MRE11 recruitment to represent functional assembly of the DNA end resection machinery. We agree that MRN loading might not necessarily correlate with long-range resection, however, we also show that formation of RAD51 foci is unaltered in the RAD54L2 knockout, further suggesting that resection is proceeding

normally. Our summary statements on lines 266-270 and lines 306-307 reflect both assays, not only MRE11 foci formation. We concluded that RAD54L2 likely functions downstream of MRE11 and RAD51.

- The discussion would benefit from addressing the observed differences between the lack of hypersensitivity in RAD54L2 KO cells upon HU treatment observed in other studies and the repair phenotypes described. Even without performing sensitivity assays, speculating on the potential implications of these observations would enrich the narrative.

Since we did not directly assess the HU sensitivity of the RAD54L2 KO cells, we prefer to not speculate on this point, although we are of course aware that lack of HU sensitivity of RAD54L2-deficient cells has been reported.

- A critical requirement for any scientific publication is the inclusion of at least $n=3$ biological replicates for all experiments, regardless of how strong the observed phenotype may appear. Exceptions can be made for screening experiments, provided they are followed by subsequent validation experiments, which should also meet the $n=3$ criteria. In this reviewer's opinion, meeting this condition is essential for the manuscript to be considered for publication in EMBO Reports, but it may be the journal/editor's discretion whether $n=2$ experiments are sufficient. This recommendation applies to Figures 3C, 3D, 3E, and 6B. Additionally, proper statistical analyses are required to support the conclusions drawn.

If one considers the work as a whole, we have shown increased SCE's in RAD54L2 knockdown (Figure 3C, $n=2$), RAD54L2 knockout (Figure 3E, $n=2$ for each of two independent knockout clones), and a second RAD54L2 knockout experiment (Figure 6B, $n=2$). This is much stronger evidence than would be provided by a single experiment with $n=3$. Similarly, the non-crossover experiment in Figure 3D (RAD54L2 knockdown, $n=2$) is followed by experiments with RAD54L2 knockouts (Figure 3F, $n=3$ for each of two independent knockout clones). While we agree that $n=3$ is popular in bio-sciences, it represents quite low statistical power, and $n=2$ is not meaningless as a measure of reproducibility of a phenomenon.

We have added statistical support for the relevant comparisons in Figures 3F, 5B, 6A, and 6D, either on the plots or in the legends, and the accompanying Supporting Data. We originally omitted the p-values from t-tests as we thought that the large effect size and small variance were apparent from the scatter plots.

Minor Points and Corrections:

- Line 134: Figure 1A should also be referred to.

We have added callouts for Fig 1A and its source data.

- Figures 1A and 1C: These panels present overlapping information and should be consolidated into a single panel for clarity.

It is true that Fig 1C is a visualization of a subset of data presented in Fig 1A, but because it incorporates additional data from BioGRID and presents a network view of the BTRR proximal proteome, we prefer to leave it as a separate panel.

- Figure 1F: Overexpression of RAD54L2 in siBLM-deficient cells was suggested by this reviewer. The authors faced challenges in recovering cells after transfection with siBLM, RAD54L2 plasmid and I-SceI plasmid simultaneously. A more practical approach would have been to first generate a cell line stably expressing RAD54L2 (e.g., via a lentiviral vector) before introducing additional experimental conditions. This strategy would have simplified the system.

Indeed it could have. It is unfortunate that we weren't able to complete that experiment.

- Lines 162-164: as far as I understand, in Fig1A/2B, PML is shown to be detected in BirA-BLM (N-terminus tagged), therefore I do not think that they authors can claim "One exception was PML, which was detected with BirA*-TOP3A, BirA*-RMI1, and BLM-mT, suggesting that PML is not in close proximity to the BLM amino terminus".

We made a poor inference here, as the reviewer notes. We have deleted those lines, and we thank the reviewer.

- Figures 2A, 2B, and 4D: "proteins identified with {greater than or equal to}2 baits are shown" should be specified, as stated in Figure 1A, to maintain the clarity.

In Fig 2A the network shows all proteins identified with BLM-mT and mT-BLM, not just those identified with ≥ 2 baits. In Fig 2B, we now have added text to the figure to indicate the proteins identified with ≥ 2 baits, and those that are unique to each bait. In Fig 4D we now have added text to the figure to indicate that the proteins identified with ≥ 2 baits are shown.

- Line 191: Statistical significance cannot be claimed with $n=1$; cannot be stated as "resulted in a statistically supported increase".

In this instance we were comparing the means of the number of SCEs per mitosis, and so $n>1$. For a similar analysis, please see Fig 7D of PMID 35115399. We have removed the statement to avoid confusion, and simply state that 13 of 18 knockdowns resulted in a greater than 2-fold increase in SCEs (lines 189-190).

- Figure EV4A: The loading control is not optimal. Since the authors are also showing the TIDE analysis for these clones, I find this a minor point; but from this blot alone one cannot conclude if the KOs are reliable.

The relevant knockout lines in EV4A are clones 1 and 2, both of which have a clear GAPDH band visible, supporting the inference that the RAD54L2 protein is no longer expressed. The data from the blot plus the TIDE analysis are consistent with both lines having a disruption of *RAD54L2*.

- Figures 4B and 4C: Ideally, an empty mt-FLAG vector control should be included to confirm interaction specificity. However, given the corroborating data between experiments, this is not critical.

We agree that an empty vector control would further strengthen the result, and that we present corroborating data to support our inference.

- Line 242: Refer to Figure 4D, not Figure 4C.

- Line 245: Refer to Figure 4D, not Figure 4C.
- Line 248: Refer to Figure 4E, not Figure 4D.
- Line 251: Refer to Figure 4E, not Figure 4D.

All callouts have been corrected.

- HU treatment details: Time and concentration for HU treatment should be included in the Results section of the manuscript or in the figure legend.

Thank you for noting our omission. The HU conditions were noted in the Methods section and are now included in the legends for Figs 5 and 6.

- Figure EV5D legend: The legend should state "As in panel C" instead of "As in panel D."

Corrected—thanks.

- Figures 6C and 6D: Since these panels derive from the same experiment, they should be presented together in the same panel.

We revised the figure legend to indicate that 6C shows representative images for the quantification in 6D. We have left them as separate panels to keep the legend and the callouts clear.

- Line 303: The data do not conclusively determine whether RAD54L2 is required for processing ultra-fine bridges or preventing their formation. The possibility of defects in decatenation by TOP2 cannot be excluded, particularly since no HU treatment was applied in this experiment. Including data on the combined effect of siBLM and RAD54L2 KO would clarify this point. But at least "processing" should be changed to "avoiding accumulation".

We agree that additional evidence would be important to distinguish a role in processing vs a role in preventing UFB formation. We have modified the text (now on line 299) accordingly.

Referee #3:

The authors addressed properly all my concerns

Grant Brown
University of Toronto
Department of Biochemistry
160 College Street
CCBR Room 1206
Toronto, ON M5S 3E1
Canada

Dear Dr. Brown,

I am very pleased to accept your manuscript for publication in the next available issue of EMBO reports. Thank you for your contribution to our journal.
